# Timed daily exercise remodels circadian rhythms in mice

Alun Thomas Lloyd Hughes [1,2], Rayna Eve Samuels[1,6], Beatriz Baño-Otálora [1,6],
Mino David Charles Belle [1,3], Sven Wegner[1], Clare Guilding[1,4], Rebecca Catrin Northeast[1],
Andrew Stewart Irvine Loudon[1], John Gigg [1] & Hugh David Piggins [1,5 ✉]

Regular exercise is important for physical and mental health. An underexplored and intriguing property of exercise is its actions on the body's 24 h or circadian rhythms. Molecular clock cells in the brain's suprachiasmatic nuclei (SCN) use electrical and chemical signals to orchestrate their activity and convey time of day information to the rest of the brain and body. To date, the long-lasting effects of regular physical exercise on SCN clock cell coordination and communication remain unresolved. Utilizing mouse models in which SCN intercellular neuropeptide signaling is impaired as well as those with intact SCN neurochemical signaling, we examined how daily scheduled voluntary exercise (SVE) influenced behavioral rhythms and SCN molecular and neuronal activities. We show that in mice with disrupted neuropeptide signaling, SVE promotes SCN clock cell synchrony and robust 24 h rhythms in behavior. Interestingly, in both intact and neuropeptide signaling deficient animals, SVE reduces SCN neural activity and alters GABAergic signaling. These findings illustrate the potential utility of regular exercise as a long-lasting and effective non-invasive intervention in the elderly or mentally ill where circadian rhythms can be blunted and poorly aligned to the external world.

[1] Faculty of Biology, Medicine and Health, The University of Manchester, Manchester, UK. [2]Present address: School of Biological and Environmental Sciences, Liverpool John Moores University, Liverpool, UK. [3]Present address: University of Exeter Medical School, Exeter, UK. [4]Present address: School of Medical Education, Newcastle University, Newcastle, UK. [5]Present address: School of Physiology, Pharmacology, and Neuroscience, University of Bristol, Bristol, UK. [6]These authors contributed equally: Rayna Eve Samuels, Beatriz Baño-Otálora. ✉email: Hugh.piggins@bristol.ac.uk

Daily rhythms in physiology and behavior are controlled by coordinated activity of clock cells in the brain's suprachiasmatic nuclei (SCN)[1]. The molecular clock within these cells enables them to function as autonomous oscillators, each with its own circadian period[2]. In SCN neurons, the molecular clock drives 24 h rhythms in electrical activity, with higher frequencies of spike firing during the day than the night[3,4]. This daily rhythm in neuronal activity facilitates intercellular signaling and synchrony among autonomous SCN cellular oscillators, which is paramount for optimal circadian function[5,6]. Two key SCN neurochemicals are the neuropeptide vasoactive intestinal polypeptide (VIP), acting via its cognate $VPAC_2$ receptor[7,8], and GABA signaling via the $GABA_A$ receptor[9–12]. In adult mice lacking VIP ($Vip^{-/-}$) or $VPAC_2$ receptor ($Vipr2^{-/-}$), SCN molecular clock and neuronal rhythms are blunted and temporally disorganized[7,13–15], while knockout of the vesicular GABA transporter ($VGAT^{-/-}$) alters burst spiking of SCN neurons but does not affect molecular clock rhythms[16].

Environmental light and internal arousal are key cues (or Zeitgebers) that synchronize the SCN with the 24 h external world[17,18]. Mice lacking VIP or $VPAC_2$ receptor do not express endogenous 24 h rhythms in behavior[13,19,20], while targeted deficiency in SCN VGAT expression attenuates mouse locomotor rhythms without altering their ~24 h period[16]. Intriguingly, sustained 24 h rhythms in behavior can be restored in $Vip^{-/-}$ and $Vipr2^{-/-}$ animals by the arousal-related cue of daily scheduled voluntary exercise (SVE)[21]. However, it is unknown whether SVE boosts SCN molecular clock rhythms and neuronal activity in neuropeptide signaling-deficient mice. Further, evidence indicates that blockade of GABA–$GABA_A$ receptor signaling in the SCN can synchronize molecular clock rhythms in the SCN of $Vip^{-/-}$ mice[10], but whether GABA signaling is altered in the SCN as a consequence of SVE is unresolved. Here we show that, following timed physical exercise, inhibitory and cell-coupling opposition actions of GABA signaling are differentially altered in the mouse SCN. In the neuropeptide signaling-deficient SCN, clock cell rhythmicity and synchrony are enhanced by this arousal cue but unaltered in the SCN of neurochemically intact mice. In contrast, neuronal activity suppressive actions of GABA are reduced in neurochemically intact SCN but remain largely unchanged in the $Vipr2^{-/-}$ SCN. A reduction in inhibitory GABA signaling is typically associated with increased firing rate, but unexpectedly, spiking in both $VPAC_2$-deficient and intact SCN is reduced following SVE. This indicates that, in the $Vipr2^{-/-}$ SCN, SVE downregulates the coupling opposing actions of GABA without alleviating its suppression of neuronal activity. These actions of timed arousal are accompanied by restoration of 24 h behavioral rhythms, despite the absence of a key SCN-synchronizing intercellular signal.

This research raises the possibility that a non-invasive intervention such as timed physical exercise may provide a mechanism to alleviate circadian misalignment[22] and be useful in the treatment of conditions associated with weakened biological timekeeping, such as bipolar disorder and the plethora of negative health indications related to shift-work[23–25].

## Results

### SVE promotes persistent ~24 h behavioral rhythms and restructures SCN temporal architecture.
As we and others have previously shown, mice deficient in $VPAC_2$ ($Vipr2^{-/-}$) or VIP ($Vip^{-/-}$) exhibit reduced behavioral rhythmicity when transferred from light:dark (LD) conditions to constant darkness (DD). Rhythmic individuals express aberrant behavioral rhythms with a shortened period of ~22.5 h and reduced rhythm strength, generated out of phase with a prior LD cycle. Such free-running rhythms differ significantly from those of wild-type (WT) mice (period typically ~23.3–24.4 h)[15,19,21,26,27]. Initial behavioral rhythms of WT, $Vipr2^{-/-}$, and $Vip^{-/-}$ mice here replicate this (Figs. 1 and 2, Figs. S1 and S3, and Table 1); thus, on transfer to constant dark, WT mice initiate their behavioral rhythm some 15–40 min in advance of the previous lights-off time, whereas for $Vip^{-/-}$ and $Vipr2^{-/-}$ animals this occurs 6–10 h in advance of this time (Fig. 1 and Fig. S1e). However, following 3 weeks of timed wheel-running (6 h/day; SVE), ~70% of $Vipr2^{-/-}$ and ~40% of $Vip^{-/-}$ mice (both significantly greater than pre-SVE proportions) sustain behavioral rhythms with periods in the typical ~24 h range expressed by WT mice. These post-SVE rhythms are much more closely aligned to the previously scheduled opportunity to exercise (0–2 h in advance of the prior onset of SVE) and, in the case of $Vipr2^{-/-}$ mice, exhibit significantly increased rhythm strength (Fig. 1, Fig. S1, and Table 1 and see ref. [21]). Concordant with our previous research[21], following this 3-week regimen, no long-term effect post-SVE is seen on the behavioral period of neurochemically intact WT mice. With longer durations, WT mice do entrain stably to SVE (Fig. 1b and Fig. S1f), with a corresponding change in post-SVE free-running period[21], in agreement with previous descriptions of the effects of extended scheduled exercise[28–31]. The phasing of this entrained rhythm is different to that of $Vip^{-/-}$ and $Vipr2^{-/-}$ animals as the onset of the WT mouse rhythm is ~8 h in advance of the opportunity to exercise (Fig. 1b and Fig. S1f). These findings reinforce our earlier published observations that, when free-running in constant dark, $Vipr2^{-/-}$ (and to a lesser extent, $Vip^{-/-}$) but not WT mice rapidly entrain to timed wheel-running.

We next assessed whether and how $Vipr2^{-/-}$ animals re-entrain behavioral rhythms to shifts in the scheduled exercise Zeitgeber by both advancing and delaying the phase of SVE by 8 h. $Vipr2^{-/-}$ mice typically resynchronized rapidly after a shift in the timing of SVE, and in some instances we saw evidence of gradual (transient) shifts in the drinking activity during re-entrainment, consistent with true entrainment to SVE rather than passive synchronization (Fig. S2a, b).

As indicated above, the entraining actions of a Zeitgeber are determined by its parameters, such as duration. To test whether short-term exposure to timed exercise can alter behavioral rhythms in mice with deficient neuropeptide signaling, we assessed responses both to reducing the number of days under SVE (8 days of SVE for 6 h per day) or the number of SVE hours per day (21 days of SVE for 1 h per day). Compared to $Vipr2^{-/-}$ animals exposed to 3 weeks of 6 h/day SVE (above), a markedly reduced proportion of $Vipr2^{-/-}$ mice exposed to an 8-day SVE protocol exhibited ~24 h rhythmicity (~20% vs. ~70%), with a moderate, but significant, increase in period towards 24 h (Fig. 2a, b vs. Fig. 1f; also see Tables 1 and 2). WT mice did not show these long-term alterations post-$SVE_{8-day}$ and instead maintained a typical ~23.6 h free-running rhythm (Fig. 2a and Table 2). Moreover, a group of $Vipr2^{-/-}$ mice exposed to 21 days of SVE for 1 h per day synchronized with this stimulus, though only one individual exhibited robust ~24 h rhythms in post-$SVE_{1h}$ free-running conditions, with most failing to display identifiable behavioral rhythms following this exercise regimen (Fig. 2c). Thus, SVE "dose-dependently" promotes persistent ~24 h behavioral rhythms in mice with disrupted VIP–$VPAC_2$ signaling.

The above experiments reveal considerable plasticity in the circadian system of neuropeptide signaling-deficient animals and we next assessed whether a non-24 h cycle of timed exercise could promote non-24 h rhythms in $Vipr2^{-/-}$ mice. To do this, we exposed a cohort of $Vipr2^{-/-}$ mice to a 3-week SVE protocol with a 25 h period ($SVE_{25h}$). All animals adjusted rapidly and appeared to synchronize to this non-24 h SVE protocol, but few stably

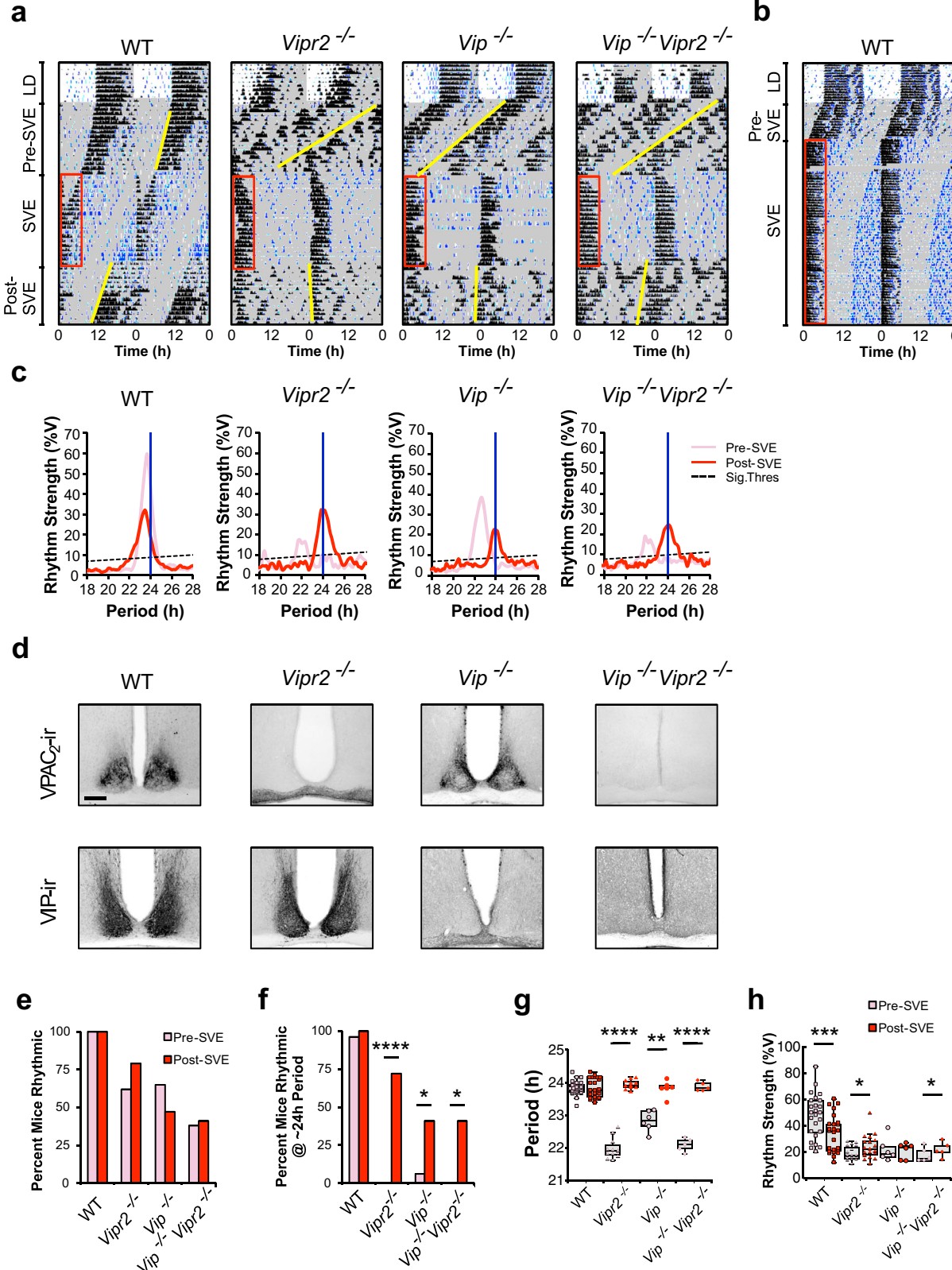

expressed ~25 h period behavioral rhythms post-SVE$_{25h}$, and the majority of $Vipr2^{-/-}$ mice that were rhythmic post-SVE$_{25h}$ expressed ~24 h rhythms in behavior (Fig. 2d–f and Table 3). Therefore, although $Vipr2^{-/-}$ animals can adapt to SVE$_{25h}$, this is not robustly maintained when the regimen is completed, and behavior instead reverts to near 24 h rhythmicity.

For the SCN to drive rhythms in behavior in constant conditions, its thousands of cell autonomous clocks must be appropriately synchronized[14]. In $Vip^{-/-}$ and $Vipr2^{-/-}$ mice, such synchrony in SCN cells is severely compromised[14,15,27,32,33]. To probe how the organization of SCN clock cell rhythms may change in response to SVE, we used time-lapse confocal

**Fig. 1 Scheduled locomotor exercise promotes ~24 h behavioral rhythms in mice with disrupted VIP–VPAC$_2$ signaling. a** Representative double-plotted actograms showing locomotor (black) and drinking (blue) activity for mice undergoing 3 weeks of scheduled voluntary exercise (SVE; $n = 25$, 39, 17 and 22, respectively). Shaded areas represent darkness. Red boxes indicate time of wheel availability during SVE and yellow lines mark onsets of activity pre- and post-SVE. **b** WT mice stably entrained to longer durations of SVE (also see Fig. S1f). **c** Chi$^2$ periodograms showing dominant circadian period and rhythm strength of wheel-running activity pre-SVE (pink/light) and post-SVE (red/dark). Diagonal broken lines indicate significance threshold at $p = 0.0001$. Vertical blue lines indicate 24 h period for reference. **d** VIP and VPAC$_2$ immunoreactivity in the SCN of wild-type (WT), $Vipr2^{-/-}$, $Vip^{-/-}$, and $Vip^{-/-}Vipr2^{-/-}$ mice (scale bar: 200 μm). **e**, **f** Bar charts showing the percentage of rhythmic mice (**e**) and percentage of rhythmic mice with ~24 h period (**f**). **g**, **h** Dot plots overlaid on box plots show the period of wheel-running activity (**g**) and rhythm strength of wheel-running activity (**h**). Dots are the individual data points used in statistical analysis. Gray shaded boxes represent the interquartile distance between the upper and lower quartile with the median plotted as a horizontal line within the box. Whiskers in **g**, **h** depict the lower quartile − 1.5 × interquartile distance and upper quartile + 1.5 × interquartile distance (only individuals rhythmic both pre- and post-SVE were included in paired $t$ tests due to the requirements of this repeated-measures assessment; $n = 25$ of 25 WT, 19 of 39 $Vipr2^{-/-}$, 6 of 17 $Vip^{-/-}$, 5 of 22 $Vip^{-/-}Vipr2^{-/-}$). Versions of **g**, **h** showing all data points pre- and post-SVE can be seen in Figs. S1g, h. Bar chart/dot shading for **f–h** is as shown in **e**. SVE significantly increased the proportion of $Vipr2^{-/-}$, $Vip^{-/-}$, and $Vip^{-/-}Vipr2^{-/-}$ mice rhythmic with an ~24 h period ($p < 0.00001$, $p = 0.0195$, and $p = 0.035$, respectively; McNemar's test). Such mice do not spontaneously generate ~24 h behavioral rhythms in extended DD in the absence of SVE (Fig. S1a, b). The period of post-SVE locomotor behavior was significantly lengthened in $Vipr2^{-/-}$, $Vip^{-/-}$, and $Vip^{-/-}Vipr2^{-/-}$ mice ($p < 0.00001$, $p = 0.0028$, and $p < 0.00001$, respectively; paired $t$ tests) and rhythm strength was significantly increased in $Vipr2^{-/-}$ and $Vip^{-/-}Vipr2^{-/-}$ mice (both $p < 0.05$); paired $t$ tests). The phase angles of entrainment were significantly different between "LD to pre-SVE (DD1)" and "SVE to post-SVE (DD2)" for $Vipr2^{-/-}$ and $Vipr2^{-/-}Vip^{-/-}$ ($p < 0.0001$) and $Vip^{-/-}$ ($p < 0.005$) mice, and there was a significantly different phase relationship between these transitions for WT mice ($p < 0.0001$; all paired $t$ tests; also see Fig. S1e). *$p < 0.05$; **$p < 0.01$; ***$p < 0.00005$; ****$p < 0.00001$. Also see Fig. S1.

microscopy[34] to image expression of a fluorescent reporter of the core clock gene *Per1* at single-cell level in SCN cultures from post-SVE WT and $Vipr2^{-/-}$ mice as well as non-SVE controls (Fig. 3). The reporter strains used here exhibited behavioral responses to SVE that were indistinguishable from those of non-reporter mice (Fig. S3a, b). Current evidence suggests that the dorsal (dSCN) subregion functions as the main period generating subregion[35,36], while the ventral (vSCN) subregion acts to integrate entrainment cues[37]. Therefore, we assessed *Per1*::eGFP expression in brain slices containing both dSCN and vSCN but found no evidence for subregional differences in the impact of SVE on clock gene synchrony and rhythmicity in WT and $Vipr2^{-/-}$ SCN (Fig. S3c–e). This suggests that both pacemaking and cue integration processes in the SCN are similarly influenced by SVE. Consistent with previous studies, a high proportion of cells in the SCN of WT mice were rhythmic, with well-synchronized oscillations, and SVE did not alter this (Fig. 3). In contrast, the proportion of rhythmic cells and their synchrony were both low in non-SVE $Vipr2^{-/-}$ SCNs (Fig. 3). Notably, SVE significantly increased the proportion of rhythmic $Vipr2^{-/-}$ SCN cells and improved cellular synchrony (Fig. 3b, c). Further, cells in non-SVE $Vipr2^{-/-}$ SCNs exhibited an abnormally wide range of periods compared to WT SCNs (period variability defined as the standard deviation of mean period for all rhythmic cells within a single slice; Fig. 3c). Following SVE, cellular periods in $Vipr2^{-/-}$ SCNs were stabilized such that period variability was significantly reduced to within a range similar to that of WTs (Fig. 3c). Consistent with rescued behavioral rhythms, these data define SVE-mediated promotion, stabilization, and resynchronization of SCN oscillator cell rhythmicity in the absence of coherent intercellular communication via VIP–VPAC$_2$ receptor signaling.

**SVE alters spontaneous electrical and GABAergic activity in the SCN.** Our observation that some animals with deficient neuropeptide signaling can sustain short-period rhythms and that SVE can restore 24 h rhythms in some of these mice indicates that other neurochemical signals contribute to rhythmicity in these animals. Since $Vip^{-/-}$ mice express VPAC$_2$ receptor and $Vipr2^{-/-}$ mice express VIP (Fig. 1d), signaling through an unidentified VPAC$_2$ agonist and VIP target, respectively, may underpin the residual rhythmicity and responsiveness to SVE in these mice. We generated mice completely deficient in this signaling system ($Vip^{-/-}Vipr2^{-/-}$; Fig. 1d) to determine whether

this would wholly abolish the 24 h rhythm-promoting actions of SVE. When transferred from LD conditions to DD, no $Vip^{-/-}Vipr2^{-/-}$ mice spontaneously exhibited ~24 h behavioral rhythms, but following 3 weeks of SVE, ~40% of $Vip^{-/-}Vipr2^{-/-}$ mice showed ~24 h behavioral rhythms (Fig. 1, Fig. S1, and Table 1). This indicates that signals largely independent of VIP–VPAC$_2$ promote circadian rhythms in these mice.

Action potentials are a key output of neural circuits and SCN neuronal firing is used to communicate circadian timekeeping information to the rest of the brain[38–40]. To assess how baseline firing was altered by the absence of VPAC$_2$ receptor signaling, we used a microelectrode array (MEA) platform to record and compare spontaneous multi-unit spiking activity in the dSCN and vSCN of brain slices from WT and $Vipr2^{-/-}$ mice (Fig. S4a). Recordings of explants from free-running non-SVE controls were initiated ~1 h after behavioral onset. For scheduled exercise mice, since the onset of behavioral rhythms in $Vipr2^{-/-}$ animals was approximately synchronized with the time that the wheel was made available, recordings for post-SVE $Vipr2^{-/-}$ animals were initiated ~1 h after the onset of wheel availability. However, because WT mice under scheduled exercise align and initiate their behavioral activity ~8 h in advance of wheel availability, post-scheduled exercise WT animals were culled in two separate groups, SVE(1) and SVE(2), to control for wheel availability and endogenous behavioral onset, respectively (Fig. S5). Post-scheduled exercise, WT mice in the SVE(1) group were sampled ~1 h after the onset of wheel availability and those in the SVE(2) group ~1 h after the onset of endogenous behavioral (drinking) activity. These two WT groups were entrained to the same SVE Zeitgeber and differed only in the phase of cull relative to behavioral onset and wheel availability. Following SVE, firing rate in the dSCN of $Vipr2^{-/-}$ mice was significantly reduced but not altered in the WT dSCN in either the SVE(1) or SVE(2) groups (Fig. 4a–e). This suggests that VPAC$_2$ signaling in WT animals may confer robustness that prevents an SVE-mediated decrease in spiking activity in the dorsal SCN. In non-SVE animals, baseline spiking rate in the whole dSCN did not differ significantly between $Vipr2^{-/-}$ and WT mice (Fig. 4a–e and Table S1), though heatmap data suggest the existence of a small cluster at the dorsal extreme of the SCN where $Vipr2^{-/-}$ firing rate is greater than that of WTs (Fig. 4f).

In the vSCN, spontaneous firing frequency for non-SVE animals was lower in $Vipr2^{-/-}$ compared with WT mice

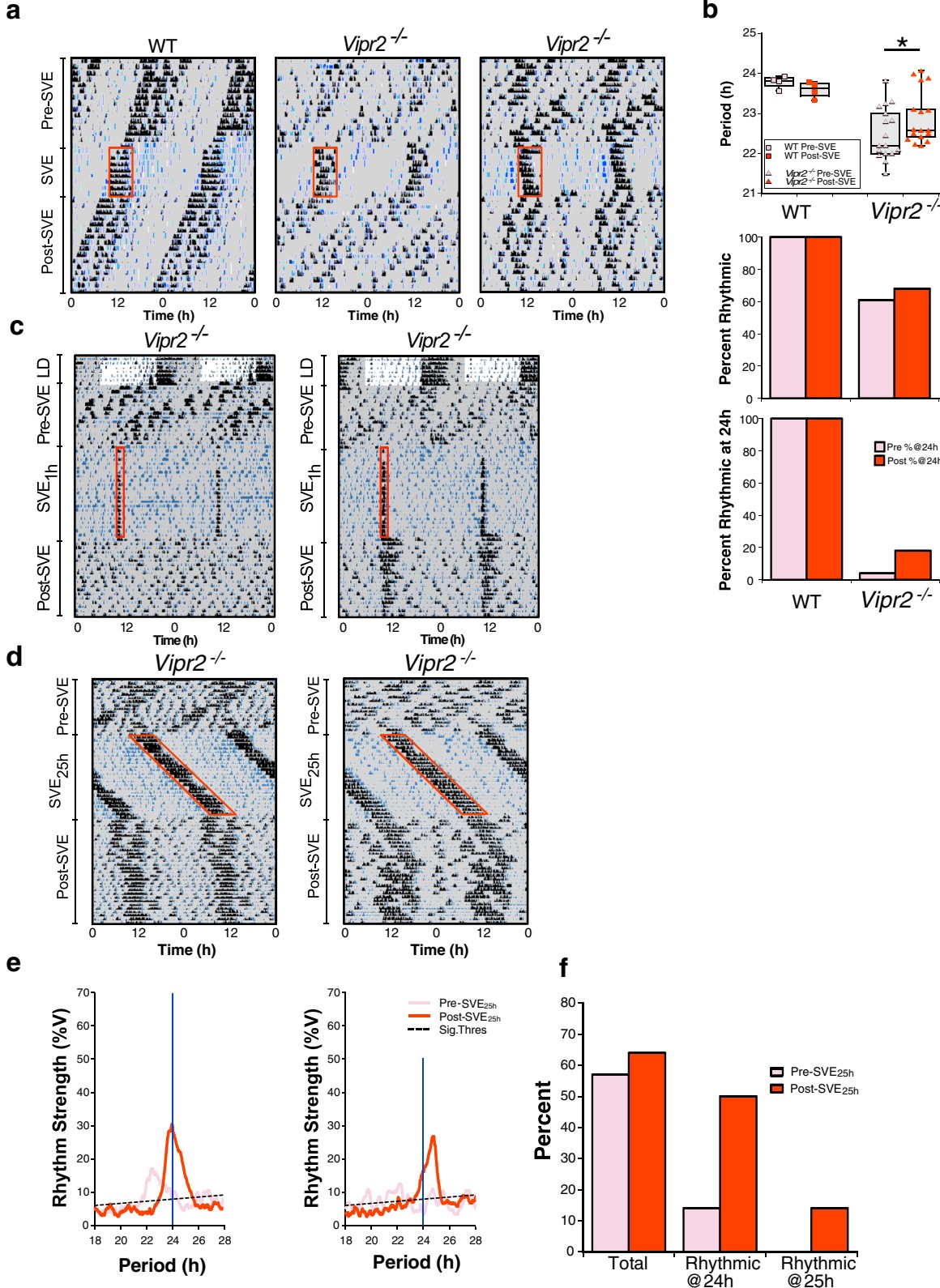

(Fig. 4c–f and Table S1). This is consistent with previous reports of the reduced firing activity of adult SCN neurons in $Vipr2^{-/-}$ mice[7,27]. The already low spontaneous firing of $Vipr2^{-/-}$ vSCN neurons was not significantly altered by SVE. In WT mice, spontaneous firing tended to be lower in animals exposed to timed wheel-running but only significantly so for SVE(1) mice (Fig. 4c–e and Table S1). This indicates that exposure to

scheduled wheel-running reduces firing rate in the dSCN but not vSCN of $Vipr2^{-/-}$ mice, while in WT animals, timed wheel-running tends to reduce firing in the vSCN.

GABA is a ubiquitous neurotransmitter in the SCN[41] and GABA signaling via the $GABA_A$ receptor can oppose coupling among SCN neurons[10,42]. Further, similar to many brain areas, SCN GABA signaling is plastic[43,44] and can be altered by varying

**Fig. 2 Neuropeptide signaling-deficient mice show dose-dependent responses to SVE but do not typically express ~25 h rhythms following an SVE Zeitgeber with a 25 h period. a** Representative double-plotted actograms showing locomotor activity (black) and drinking activity (blue) for WT and *Vipr2*$^{-/-}$ mice ($n = 4$ and 28, respectively) undergoing an 8-day 24 h SVE protocol. Shaded areas represent darkness. Red boxes indicate time of wheel availability during SVE (6 h). **b** Dot plot overlaid on box plot depicts the period of wheel-running activity, while bar charts show the percentage of mice rhythmic (in total) and percentage of mice rhythmic with ~24 h period pre- and post-SVE. Gray shaded boxes represent the interquartile distance between the upper and lower quartile with the median plotted as a horizontal line within the box. Whiskers in **b** depict the lower quartile − 1.5 × interquartile distance and upper quartile + 1.5 × interquartile distance, while the individual data point symbols show only points that contributed to statistical assessment (only individuals rhythmic both pre- and post-SVE were included in paired *t* tests due to the requirements of this repeated-measures assessment; $n = 4$ of 4 WT, 17 of 28 *Vipr2*$^{-/-}$). A version of this panel showing all data points pre- and post-SVE can be seen in Fig. S2c. Bar/dot shading for upper and middle panels of **b** is as shown in the lower panel. The increase from 4 to 18% of the *Vipr2*$^{-/-}$ population ($n = 28$ total) expressing ~24 h rhythms following 8 days of SVE is not significant. However, the mean period of post-8-day SVE *Vipr2*$^{-/-}$ mice was significantly lengthened (22.94 ± 0.16 h post-SVE vs. 22.5 ± 0.03 h pre-SVE; $p = 0.023$; paired *t* test). *$p = 0.023$. Post 8-day SVE, WT behavioral period was significantly shorter than pre-8-day SVE, which represents the typical shortening of period associated with continued free run in constant darkness and is not a result of 8-day SVE (see Table 1). **c** Representative double-plotted actograms showing locomotor activity (black) and drinking activity (blue) for *Vipr2*$^{-/-}$ mice ($n = 8$) undergoing a 1 h per day SVE protocol (SVE$_{1h}$). Shaded areas represent darkness. Red boxes indicate time of wheel availability during SVE (1 h). One individual exhibited robust ~24 h rhythms in behavior after 1 h/day of SVE for 21 days, but the remaining 7 individuals failed to express clearly identifiable rhythms. See also Table 1. **d** Representative double-plotted actograms showing locomotor activity (black) and drinking activity (blue) for *Vipr2*$^{-/-}$ mice ($n = 14$) undergoing a 25 h SVE protocol (SVE$_{25h}$). Here the animals have the opportunity to voluntarily exercise in the running wheel for 6 h every 25 h. Shaded areas represent darkness. Red boxes indicate time of wheel availability during SVE. **e** Chi$^2$ periodograms showing dominant circadian period of running-wheel activity both pre-SVE$_{25h}$ (pink/light) and post SVE$_{25h}$ (red/dark). Diagonal broken lines indicate significance threshold at $p = 0.0001$. Vertical blue lines indicate 24 h period for visual reference. **f** Bar chart showing percentages of mice rhythmic (in total), rhythmic with ~24 h period, and rhythmic with ~25 h period, pre- and post-SVE$_{25h}$. Few *Vipr2*$^{-/-}$ mice express ~25h period behavioral rhythms post-SVE$_{25h}$, with most mice that are rhythmic expressing ~24 h rhythms in behavior. Also see Table 1.

### Table 1 Behavioral parameters for 3-week SVE experiment.

| 3-Week SVE$_{24h}$ | | WT | *Vipr2*$^{-/-}$ | *Vip*$^{-/-}$ | *Vip*$^{-/-}$*Vipr2*$^{-/-}$ |
|---|---|---|---|---|---|
| | *N* | 25 | 39 | 17 | 22 |
| % Rhythmic | Pre-SVE | 100 | 62 | 65 | 38 |
| | Post-SVE | 100 | 79 | 47 | 41 |
| | *p* (Pre vs. Post) | ns | ns | ns | ns |
| % Rhythmic @~24 h | Pre-SVE | 96 | 0 | 6 | 0 |
| | Post-SVE | 100 | 72 | 41 | 41 |
| | *p* (Pre vs. Post) | ns | $p < 0.00001$ | $p = 0.012$ | $p = 0.035$ |
| Period | Pre-SVE | 23.83 ± 0.04 | 22.34 ± 0.16 | 22.90 ± 0.17 | 22.04 ± 0.09 |
| | Post-SVE | 23.84 ± 0.06 | 24.12 ± 0.10 | 23.66 ± 0.22 | 23.90 ± 0.05 |
| | *p* (Pre vs. Post) | ns | $p < 0.00001$ | $p = 0.0028$ | $p < 0.00001$ |
| Rhythm strength (%V) | Pre-SVE | 47.2 ± 3.3 | 18.3 ± 1.1 | 17.4 ± 2.4 | 19.0 ± 3.3 |
| | Post-SVE | 33.6 ± 2.8 | 23.5 ± 1.8 | 21.7 ± 2.2 | 19.7 ± 2.2 |
| | *p* (Pre vs. Post) | $p = 0.00004$ | $p = 0.015$ | ns | $p = 0.016$ |

### Table 2 Behavioral parameters for 8-day SVE experiment.

| 8-Day SVE$_{24h}$ | | WT | *Vipr2*$^{-/-}$ |
|---|---|---|---|
| | *N* | 4 | 28 |
| % Rhythmic | Pre-SVE | 100 | 61 |
| | Post-SVE | 100 | 68 |
| | *p* (Pre vs. Post) | ns | ns |
| % Rhythmic @~24 h | Pre-SVE | 100 | 5 |
| | Post-SVE | 100 | 18 |
| | *p* (Pre vs. Post) | ns | ns |
| Period | Pre-SVE | 23.78 ± 0.08 | 22.50 ± 0.03 |
| | Post-SVE | 23.59 ± 0.10 | 22.94 ± 0.16 |
| | *p* (Pre vs. Post) | $p = 0.014$ | $p = 0.047$ |
| Rhythm strength (%V) | Pre-SVE | 49.5 ± 6.6 | 13.1 ± 0.6 |
| | Post-SVE | 44.0 ± 4.7 | 13.2 ± 1.0 |
| | *p* (Pre vs. Post) | ns | ns |

### Table 3 Behavioral parameters for 3-week SVE$_{25h}$ experiment.

| 3-Week SVE$_{25h}$ | | *Vipr2*$^{-/-}$ |
|---|---|---|
| | *N* | 14 |
| % Rhythmic | Pre-SVE | 57 |
| | Post-SVE | 64 |
| | *p* (Pre vs. Post) | ns |
| % Rhythmic @~24h | Pre-SVE | 14 |
| | Post-SVE | 50 |
| | *p* (Pre vs. Post) | ns |
| % Rhythmic @~25h | Pre-SVE | 0 |
| | Post-SVE | 14 |
| | *p* (Pre vs. Post) | ns |
| Period | Pre-SVE | 22.22 ± 0.34 |
| | Post-SVE | 24.240.12 |
| | *p* (Pre vs. Post) | $p = 0.0017$ |
| Rhythm strength (%V) | Pre-SVE | 21.7 ± 0.8 |
| | Post-SVE | 26.8 ± 1.3 |
| | *p* (Pre vs. Post) | $p = 0.018$ |

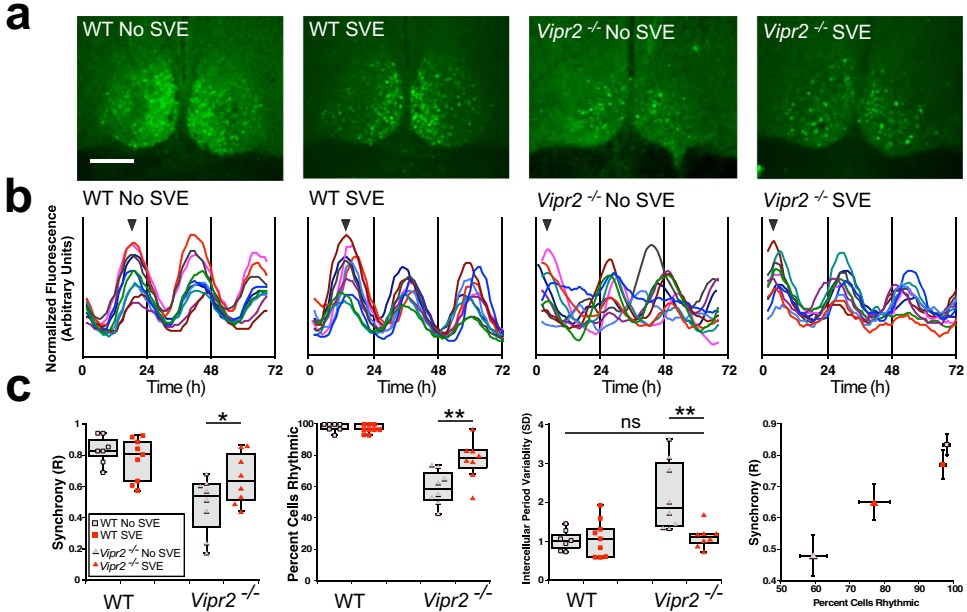

**Fig. 3 SVE significantly improves *Vipr2*$^{-/-}$ SCN temporal architecture. a** Photomicrographs of WT and *Vipr2*$^{-/-}$ SCN tissue from non-SVE and SVE mice expressing *mPer1::d2eGFP* fluorescence, resolved to the single-cell level (scale bar: 200 μm). **b** Example rhythm profiles show 10 cells each from single bilateral SCN recordings of *mPer1::d2eGFP* fluorescence. Arrowheads in **b** indicate phase of images shown in **a**. **c** SVE significantly increased the synchrony (Rayleigh *R* increased to 0.65 ± 0.06 from 0.48 ± 0.07 (mean ± SEM); *p* = 0.039) and rhythmicity (77 ± 5% vs. 59 ± 4%; *p* = 0.001) and reduced the intercellular variability (SD of intercellular periods reduced to 1.1 ± 0.1 from 2.2 ± 0.3; *p* = 0.001) of rhythms resolved at the single-cell level in *Vipr2*$^{-/-}$ SCN-containing brain slices. Correlation plot (**c**, far right panel) illustrates the relationship between cellular synchrony and percentage of cells rhythmic across SCNs from both genotypes and experimental conditions. **p* < 0.05; ***p* < 0.01. WT non-SVE: 210 cells from 7 slices; WT SVE SCN: 270 cells from 9 slices; non-SVE *Vipr2*$^{-/-}$ SCN: 190 cells, 8 slices; post-SVE *Vipr2*:$^{-/-}$ 210 cells, 8 slices. Gray shaded boxes represent the interquartile distance between the upper and lower quartile with the median plotted as a horizontal line within the box. Whiskers in **c** depict the lower quartile − 1.5 × interquartile distance and upper quartile + 1.5 × interquartile distance. Individual data points overlaid. Also see Fig. S3.

environmental signals such as daylength[9,45,46]. Indeed, experimental and in silico research indicates that plasticity in GABAergic signals can vary between dSCN and vSCN subregions[12,47]. Here we determined the contribution of GABA–GABA$_A$ receptor signaling both to non-SVE control and post-SVE spiking activity in dSCN and vSCN subregions by monitoring changes in multi-unit activity in response to the GABA$_A$ antagonist, gabazine. In animals not exposed to SVE, dSCN neurons of both genotypes increase firing rate when challenged with gabazine (Fig. 5a–d and Table S2), with a significantly larger magnitude response recorded in WT compared with *Vipr2*$^{-/-}$ animals (Fig. 5b–e). This indicates that GABA–GABA$_A$ receptor signaling plays a more prominent role in setting the spontaneous firing rate in the dSCN of WT than in *Vipr2*$^{-/-}$ mice. Following exposure to SVE, this increased spiking frequency in response to gabazine was significantly reduced in WT [in both the SVE(1) and SVE(2) groups] but not *Vipr2*$^{-/-}$ mice (Fig. 5a–d and Table S2). This indicates that timed wheel-running suppresses the inhibitory GABAergic contribution to spontaneous multi-unit firing activity in the neurochemically intact but not neuropeptide signaling-deficient dSCN (Fig. 5a–d). Thus, the decrease in *Vipr2*$^{-/-}$ dSCN spontaneous neuronal activity under scheduled voluntary activity (Fig. 4) is most likely attributable to non-GABA-dependent mechanisms.

In non-SVE control mice, gabazine evoked much larger increases in firing in the vSCN of WT compared with *Vipr2*$^{-/-}$ mice, indicating that similar to the dSCN the inhibitory GABAergic influence on vSCN spontaneous spiking activity was reduced in the *Vipr2*$^{-/-}$ mouse (Fig. 5a–e). The response to gabazine of *Vipr2*$^{-/-}$ vSCN neurons was not affected by SVE, whereas in WT mice gabazine's actions were reduced both in SVE

(1) and SVE(2) animals (Fig. 5a–d). This indicates that timed wheel-running reduces inhibitory GABA signaling in the WT vSCN, whereas in the vSCN of neuropeptide signaling-deficient mice it does not influence spontaneous firing or the inhibitory action of GABA on action potential discharge. A reduction in inhibitory GABAergic tone would be predicted to increase neural activity. However, spontaneous firing rate in the WT and *Vipr2*$^{-/-}$ vSCN is reduced following scheduled exercise, suggesting that mechanisms independent of GABA contribute to its suppression of neural activity.

**SVE improves SCN molecular rhythms.** SVE had distinct subregional effects on the neurophysiology of *Vipr2*$^{-/-}$ SCN in intact brain slices. Given the large body of evidence for different roles of the dorsal and ventral parts of the SCN, including in responding to entrainment signals[46,48,49], we made luminometric recordings of rhythms in PERIOD 2-driven luciferase (PER2::LUC) expression in microdissected dorsal-only (do) and ventral-only (vo) SCN explants (Fig. 6a–h). We used PER2::LUC animals since in pilot investigations we found that ex vivo tissue-level rhythms in molecular activities from these animals are more sustained with this passive recording configuration than with active fluorescence imaging. Luciferase-expressing strains exhibited behavioral responses to SVE that were indistinguishable from those of non-reporter mice (Fig. S3a, b). In tissue explants of SCN with disrupted VIP–VPAC$_2$ receptor signaling, PER2::LUC amplitude is low[50,51], but blockade of GABA$_A$ receptor signaling synchronizes SCN cells and increases the amplitude of PER2::LUC rhythms[10]. Therefore, we tested how timed exercise influenced these rhythms in voSCN and doSCN microdissected mini slices, as well as gabazine effects in whole SCN explants from SVE and non-SVE

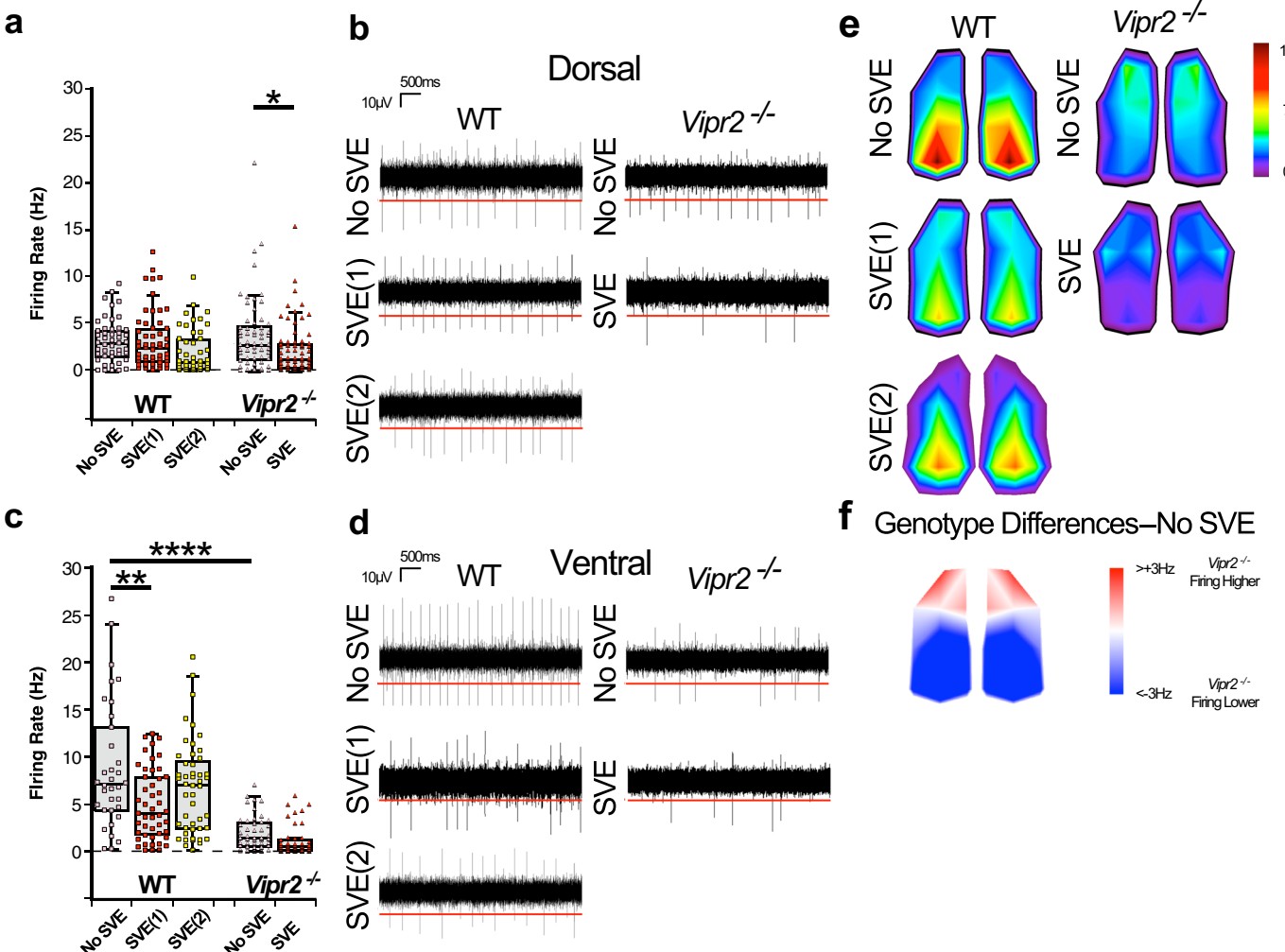

**Fig. 4 Altered spontaneous action potential firing in the SCN of WT and neuropeptide signaling-deficient mice and its manipulation by SVE.** Box plots overlaid with dot plots (**a**, **c**), example recordings (**b**, **d**), and topographical heatmaps (**e**, **f**) showing multiunit MEA recordings of spontaneous action potential firing at (~CT13 for non-SVE control mice, 1 h following onset of wheel availability/onset of activity for SVE $Vipr2^{-/-}$ animals, and either 1 h following the onset of wheel availability (SVE(1)) or onset of behavior (drinking; SVE(2)) in WT mice). Also see Fig. S5. In the dorsal SCN subregion, firing rate varied across genotypes and exercise condition (1-way ANOVA; $p < 0.0001$) (see also Table S1). In the WT SCN, scheduled exercise did not alter action potential frequency in the dorsal subregion (mean ± SEM; 3.1 ± 0.3, 3.2 ± 0.4, and 2.1 ± 0.4 Hz, respectively, for non-SVE, SVE(1), and SVE(2) ($n = 48$, 54, and 45 recording electrodes); both $p > 0.05$; **a**, **b**; Table S1). Firing rate in the $Vipr2^{-/-}$ dorsal SCN did not differ from WT mice ($p > 0.05$), but scheduled exercise reduced spontaneous action potential frequency (3.8 ± 0.5 vs. 2.3 ± 0.3 Hz; $n = 59$ and 67; $p = 0.041$ **a**, **b**). In the ventral subregion, firing rate varied across genotypes and exercise condition (1-way ANOVA; <0.0001). In the WT ventral SCN, action potential frequency was reduced by scheduled exercise in the SVE(1) condition (8.8 ± 1.1 vs. 4.9 ± 0.6 Hz; $n = 35$ and 47, $p = 0.01$) but not in the SVE(2) group (6.8 ± 0.7 Hz; $n = 48$, $p > 0.05$; **c**, **d**). In the $Vipr2^{-/-}$ ventral SCN, spontaneous firing rate was lower than WT ($p < 0.0001$), but scheduled exercise did not significantly alter firing rate (1.9 ± 0.3 vs. 1.1 ± 0.2 Hz; $n = 47$ and 45; $p > 0.05$; **c**, **d**). Heatmaps show average firing (**e**) and differences in firing between non-SVE WT and $Vipr2^{-/-}$ SCN (**f**). Horizontal red lines in **b**, **d** show detection threshold at −17 µV. Gray shaded boxes in **a**, **c** represent the interquartile distance between the upper and lower quartile with the median plotted as a horizontal line within the box. Whiskers depict the lower quartile − 1.5 × interquartile distance and upper quartile + 1.5 × interquartile distance. Individual data points are overlaid. Recordings were made from six SCN-containing brain slices from WT non-SVE mice, and seven slices each from WT SVE(1), WT SVE(2), $Vipr2^{-/-}$ non-SVE, and $Vipr2^{-/-}$ SVE mice. *$p < 0.05$; **$p < 0.01$; ****$p < 0.0001$. Also see Fig. S4. Further details of statistical outcomes are in Table S1.

mice. Notably, coincident with the changes in spontaneous action potential firing in the $Vipr2^{-/-}$ dSCN seen following SVE, the amplitude of PER2::LUC rhythms in the voSCN was increased significantly post-SVE (Fig. 6d–f). This extends the findings of an earlier investigation into the effects of SVE under LD conditions using intact SCN explants[52] and suggests that SVE promotes rhythmic activity in the $Vipr2^{-/-}$ vSCN. Coincident with the post-SVE neurophysiological changes in the WT SCN, we recorded a significant reduction in the amplitude of PER2::LUC oscillations following SVE in both doSCN and voSCN from these

neuropeptide-competent mice (Fig. 6a–f). We found no differences in period between SVE and control doSCN and voSCN explants from either WT or $Vipr2^{-/-}$ mice (Fig. 6g, h), though as previously noted, the periods of PER2::LUC expression in both genotypes were longer than 24 h[50,53]. Due to increased malleability to control treatments in doSCN and voSCN explants, we subsequently used intact SCN explants to examine the effects of GABAergic signaling inhibition on PER2::LUC oscillations. Gabazine evoked significant increases in the amplitude of PER2::LUC rhythms in intact SCN explants from non-SVE WT and

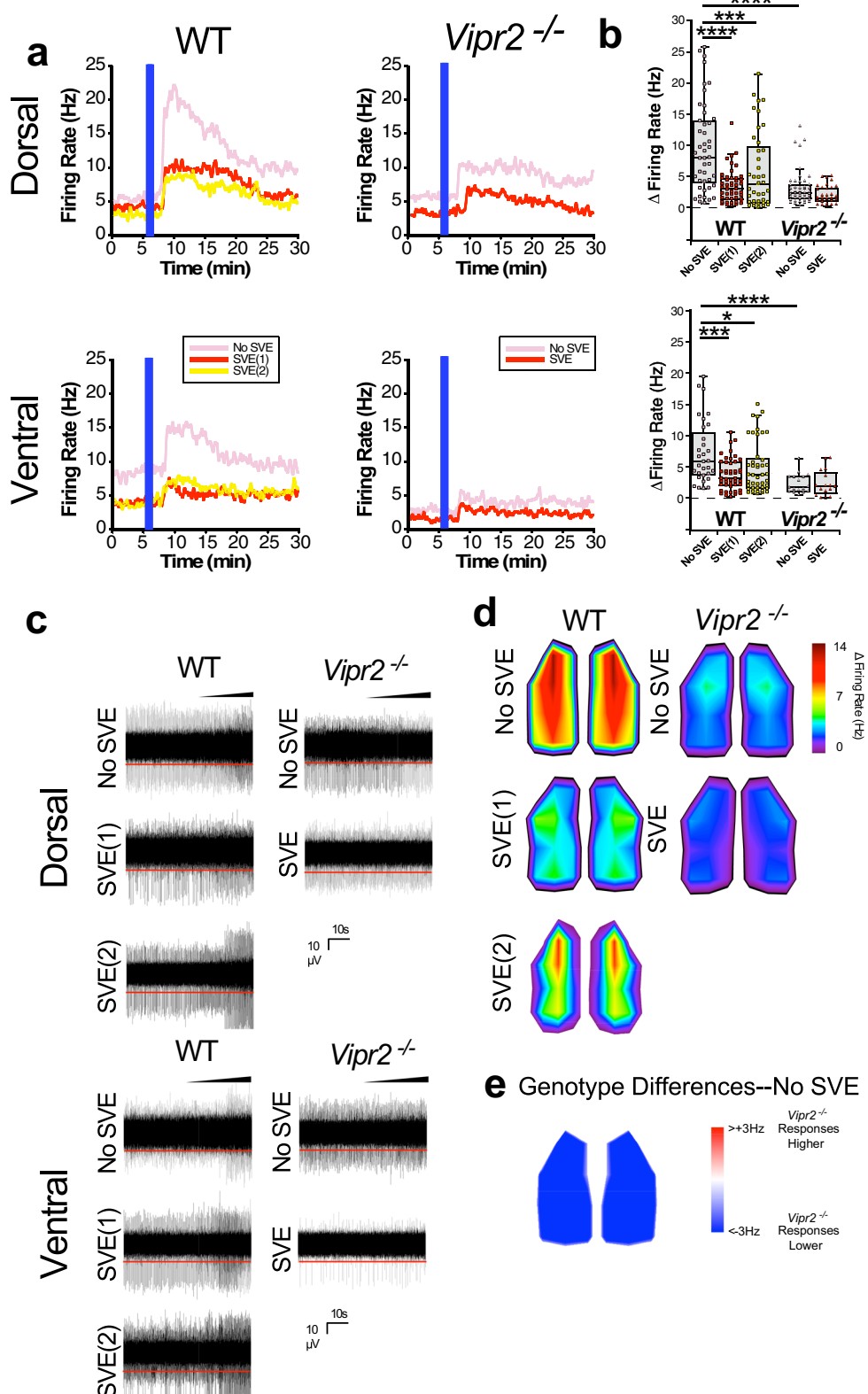

*Vipr2*$^{-/-}$ mice, but following SVE, these were abolished and reduced in WT and *Vipr2*$^{-/-}$ SCN, respectively (Fig. 7b–c). This is consistent with the reduction in GABAergic activity in WT SCN as recorded in MEA experiments following SVE (Fig. 5). Smaller amplitude changes in *Vipr2*$^{-/-}$ SCN PER2::LUC expression after SVE (Fig. 7b, c) suggest reduced GABAergic activity in the *Vipr2*$^{-/-}$ SCN following this behavioral

intervention. As no significant exercise-related changes in GABAergic signaling were detected in recordings of multi-unit activity in the *Vipr2*$^{-/-}$ SCN (Fig. 5), this indicates differential contributions of GABA–GABA$_A$ signaling to neuronal discharge and clock cell synchrony in these mice. Therefore, the alterations in molecular clock amplitude and behavioral period elicited by timed wheel-running potentially require suppression of GABA's

**Fig. 5 Altered GABAergic signaling in the SCN of WT and neuropeptide signaling-deficient mice and its manipulation by SVE. a** Example firing rate response plots, **b** box plots overlaid with dot plots, **c** example recordings, and **d**, **e** topographical heatmaps showing multiunit firing rate responses of SCN to treatment with 100 μM gabazine, recorded using MEA. In the dorsal SCN subregion, firing rate response to gabazine varied over genotype and exercise condition (1-way ANOVA; $p < 0.0001$; see also Table S2). Scheduled voluntary exercise did not significantly reduce the firing rate response of dorsal $Vipr2^{-/-}$ SCN to gabazine ($3.4 \pm 0.4$, $2.2 \pm 0.2$ Hz, non-SVE and SVE, respectively; $n = 46$ and 34 recording electrodes; $p > 0.05$) but significantly reduced the response of WT dorsal SCN neurons to GABA blockade (dorsal WT: $9.7 \pm 1.0$, $3.5 \pm 0.4$, and $6.1 \pm 1.0$ Hz, $n = 45$, 46, and 39; non-SVE, SVE(1), and SVE (2), respectively, both $p < 0.001$; Table S2). In the ventral SCN, firing rate response to gabazine varied over genotype and exercise condition (1-way ANOVA; $p < 0.0001$). Scheduled voluntary exercise significantly reduced the response to gabazine of ventral WT SCN neurons: $7.4 \pm 0.9$, $3.8 \pm 0.4$, $5.0 \pm 0.6$ Hz, $n = 31$, 46, and 44, respectively, $p < 0.01$ and $p < 0.05$) but did not alter the firing rate response of ventral $Vipr2^{-/-}$ SCN neurons to gabazine ($2.4 \pm 0.4$ vs. $2.5 \pm 0.5$ Hz; $n = 17$ and 16; $p > 0.05$; Table S2). Both dorsal and ventral $Vipr2^{-/-}$ responses to gabazine in SCNs from non-SVE animals were lower than corresponding WT values (both $p < 0.0001$). Vertical blue lines on traces in **a** indicate the time of treatment with gabazine. Gray shaded boxes in **b** represent the interquartile distance between the upper and lower quartile with the median plotted as a horizontal line within the box. Whiskers depict the lower quartile $- 1.5 \times$ interquartile distance and upper quartile $+ 1.5 \times$ interquartile distance. Individual data points are overlaid. **c** Horizontal red lines in **c** show detection threshold at $-17$ μV and sloped black bars indicate increasing concentration of gabazine in the slice chamber during gabazine wash-in. **d**, **e** Topographical heatmaps showing average changes in SCN firing in response to gabazine treatment (**d**) and average differences in response to gabazine treatment between non-SVE recordings for WT and $Vipr2^{-/-}$ SCN (**e**). Numbers of slices recorded as for Fig. 4. *$p < 0.05$; ***$p < 0.001$; ****$p < 0.0001$. Also see Fig. S4. Further details of statistical outcomes are in Table S2.

actions on SCN neuronal activity in the neurochemically intact WT mouse, whereas in the $Vipr2^{-/-}$ SCN, exercise reduces multi-unit activity via GABA-independent mechanisms but promotes clock cell synchrony through a reduction in GABAergic tone.

## Discussion

In this study, we observe that, in all three neuropeptide signaling-deficient genotypes, near 24 h rhythms can be rescued in 40–70% of animals by timed daily wheel-running. Importantly, unlike exposure to light, recurrent timed physical exercise precisely aligns the circadian system in these mice such that, on withdrawal of this arousal cue, rhythmic animals initiate behavioral activity close to the time that the opportunity for running-wheel exercise had previously been scheduled. In regard to intercellular signaling in the SCN, we find that, in the destabilized SCN of VPAC$_2$ receptor-deficient mice, the oppositional action of GABA–GABA$_A$ receptor signaling on clock cell coupling is downregulated by SVE and this associates with improved SCN cellular synchrony and stable near 24 h rhythms in behavior. This is not, however, paralleled by similar changes in the inhibitory action of GABA–GABA$_A$ receptor signaling on SCN neuronal activity. Indeed, paradoxically, SCN basal firing rate tends to be lowered, not elevated, by timed exercise in the $Vipr2^{-/-}$ mice. Interestingly, GABAergic tone is more prominent in the WT SCN, but it is not disruptive to SCN cellular synchrony or the expression of behavioral rhythms, presumably due to the overwhelming synchronizing effect of VIP–VPAC$_2$ receptor signaling. For WT mice, exercise-related reduction in this inhibitory GABA–GABA$_A$ receptor signal is without obvious consequence for SCN synchrony and behavioral rhythms, though it does reduce SCN molecular rhythm amplitude, thereby potentially contributing to the entrainment of SCN-controlled rhythms in behavior to scheduled locomotor activity. Indeed, with continued adherence to the 6 h daily exercise regimen used here for a further 2–3 weeks (7 weeks in total), WT mice do stably synchronize their behavioral rhythms to SVE[21]. Thus, with deficiencies in neuropeptide signaling, recurrent physical exercise promotes SCN cellular synchrony and expression of near 24 h rhythms of behavior through downregulating the clock cell-coupling opposing actions of intrinsic GABAergic neurotransmission within the SCN.

GABA can be inhibitory[54,55] or excitatory[56,57] in the adult SCN, and while we did not set out to define the polarity of SCN GABA signaling, our results suggest that, at the phases of the circadian cycle tested, GABA exerts a predominantly inhibitory influence. Blockade of GABA$_A$ receptors with gabazine

unequivocally activates multi-unit activity throughout the ventral and dorsal SCN subregions of WT and VPAC$_2$ receptor-deficient mice. In control animals not subject to timed wheel-running, the excitatory effects of GABA$_A$ receptor blockade were notably larger in WT compared to $Vipr2^{-/-}$ mice, indicating that basal GABAergic tone is downregulated in the neuropeptide signaling-deficient SCN, an observation consistent with previous investigations[58,59]. A contribution of excitatory responses to GABA in the SCN in general, and perhaps specifically in sculpting responses to scheduled exercise, is not precluded by our dataset, however. While our data are consistent with mainly inhibitory responses to GABA in the SCN, an as yet undetected shift in the relative balance of inhibitory and excitatory responses to GABA following schedule exercise remains possible.

Concordant with the view that clock cell coupling and synchrony in the SCN are promoted by VIP and opposed by GABA[9,10], gabazine, in the absence of SVE, elicited a pronounced elevation of PER2::LUC amplitude in the SCN of both VPAC$_2$ receptor knockout and WT mice. Further, although the $Vipr2^{-/-}$ SCN GABA–GABA$_A$ receptor signal is muted with respect to WT, $Vipr2^{-/-}$ SCN cellular synchrony is impaired, presumably due to the absence of the coupling-promoting actions of VIP–VPAC$_2$ receptor communication in opposing GABA's activity in these mice. Nonetheless, our observation of elevated molecular clock amplitude following GABA blockade is consistent with earlier investigations in which gabazine activated SCN electrical activity, boosted clock reporter rhythms[42], and promoted cellular coupling to increase SCN clock cell synchrony[10]. The reduction in GABA's cell-coupling actions following exercise, with corresponding improved synchrony of $Per1$::eGFP cells throughout the SCN of $Vipr2^{-/-}$ but not of WT mice, adds credence to this idea, further supporting the view that GABA signaling opposes coupling in the SCN[60]. These findings are consistent with the suggestion that GABA signaling functions in this way specifically when SCN steady state is destabilized through exposure to differing amounts of daily light[9] or length of days[61], perhaps representing a mechanism to aid recovery to steady state. Indeed, $Vip^{-/-}$ mice do not readily adjust SCN electrical activity or behavior to long daylengths[62].

In addition to phase-shifting and entraining circadian rhythms[21,63,64], strikingly, physical exercise can exert long-term changes in the brain, promoting synaptic plasticity in rodents[65,66], increasing GABA release[67,68], and remodeling GABA$_A$ receptor subunit expression[69]. Indeed, here we describe long-term changes in GABAergic signaling evoked in the SCN by timed wheel-running. Since VIP neurons in the SCN can release

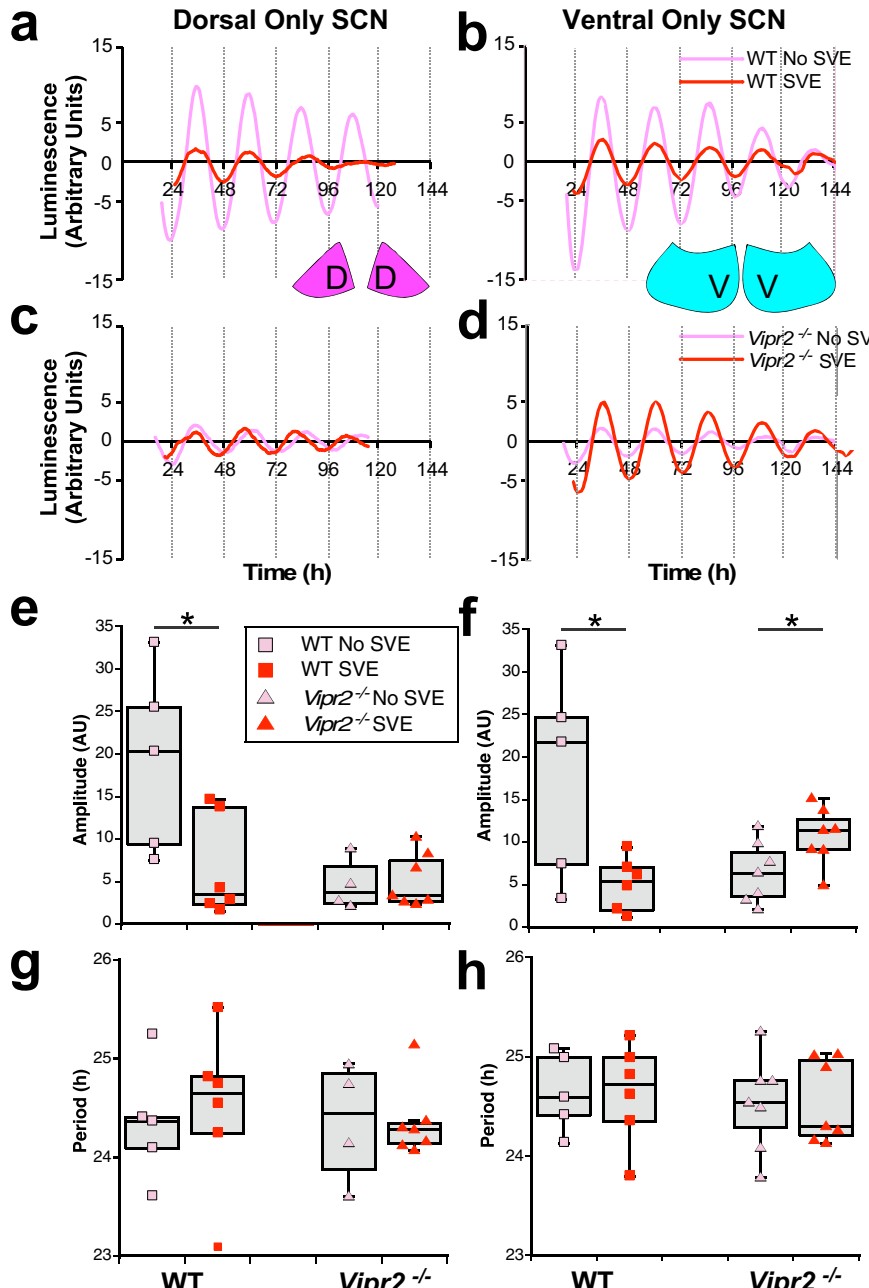

**Fig. 6 SVE improves molecular rhythms in the ventral SCN in microdissected explants from *Vipr2*⁻/⁻ mice. a–d** PER2::LUC bioluminescence amplitude was significantly reduced by SVE in the WT dorsal only SCN (doSCN; pink filled D diagram insert) and ventral only SCN (voSCN; blue filled diagram insert) microdissected mini slices (dorsal: 19.2 ± 4.8 vs. 6.5 ± 2.4 arbitrary units, *p* = 0.018; ventral: 18.0 ± 5.6 vs. 5.0 ± 1.3, *p* = 0.017) but was boosted by SVE in the *Vipr2*⁻/⁻ voSCN (6.4 ± 1.4 vs. 10.7 ± 1.3; *p* = 0.041, all *t* tests). SVE did not alter amplitude in the *Vipr2*⁻/⁻ doSCN (4.6 ± 1.5 vs. 5.1 ± 1.2; *p* > 0.05). Period (**g**, **h**) was not altered by SVE in either part of the SCN for either genotype. Gray shaded boxes in **e–h** represent the interquartile distance between the upper and lower quartile with the median plotted as a horizontal line within the box. Whiskers depict the lower quartile − 1.5 × interquartile distance and upper quartile + 1.5 × interquartile distance. Individual data points are overlaid. *\*p* = 0.05. Symbol color coding in **f–h** is as shown in **e**. **a**, **c**, **e**, **g** show data from doSCN. **b**, **d**, **f**, **h** show data from vo SCN. See also Figs. S3 and S4. D and V in diagram inserts label dorsal SCN and ventral SCN, respectively.

GABA[70], activation of the VPAC₂ receptor can stimulate presynaptic GABA release from SCN neurons[59,71], and mice with intact VIP–VPAC₂ receptor signaling entrain less readily to scheduled exercise than neuropeptide signaling-deficient strains, and functional VIP signaling in the SCN may resist the GABA remodeling influences of arousal-related feedback from timed wheel-running.

Our results also indicate that GABAergic neurotransmission, and its modulation by exercise, is complex in the SCN; alterations in these parameters vary by genotype as well as between SCN subregions, and contrary to expectation, its acute electrophysiological influences do not necessarily co-relate to its longer-term action on clock cell synchrony. Since GABA reuptake transporters[72,73] and multiple GABA_A receptor subunits are expressed in the SCN, including those underpinning extra-synaptic (tonic) as well as synaptic (phasic) actions of GABA[74–76], dissecting and identifying precisely how exercise re-organizes GABA neurotransmission in the presence or absence of

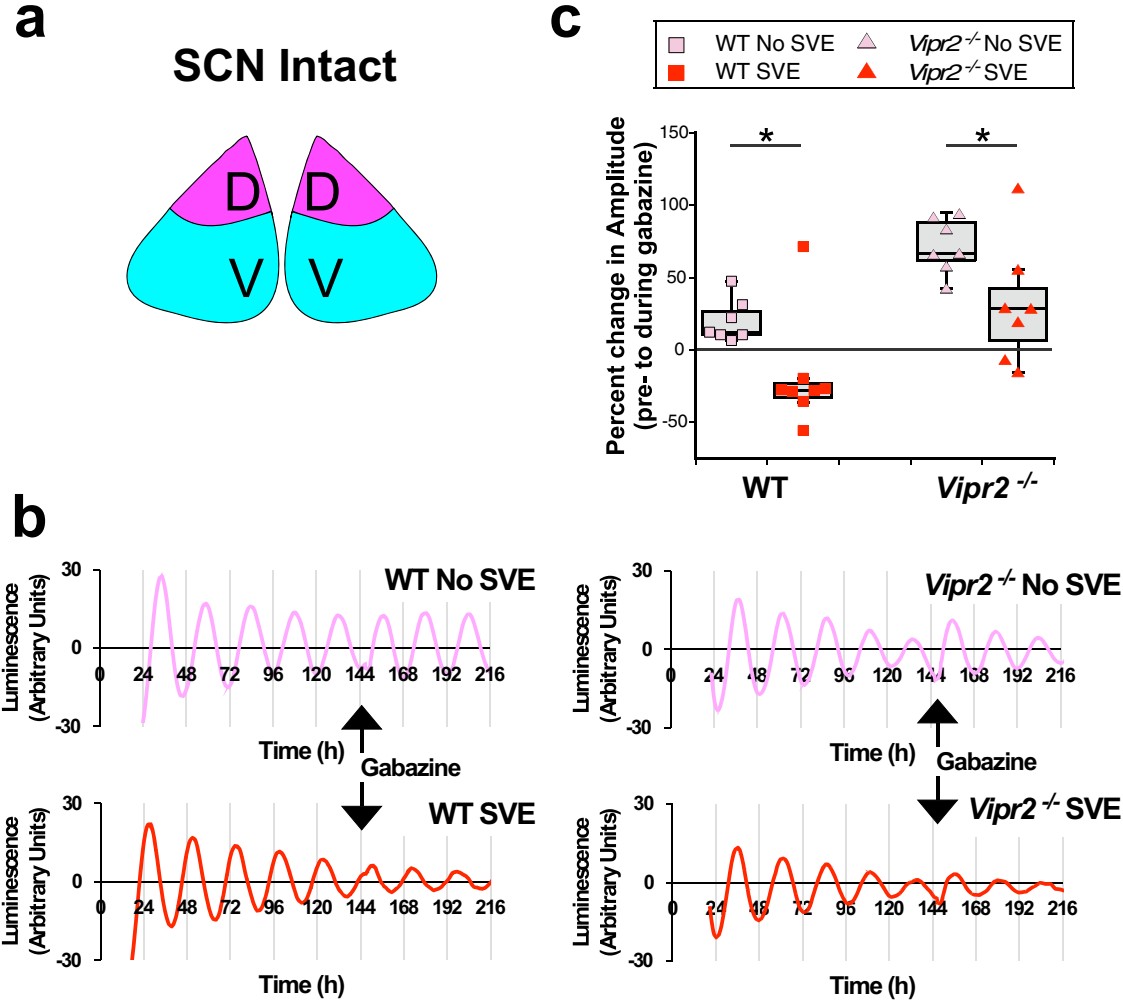

**Fig. 7 SVE reduces GABAergic influences on PER2 rhythms in the intact SCN. a** Schematic diagram of intact SCN. **b, c** Gabazine treatment evoked significant increases in PER2::LUC rhythm amplitude in intact SCN (pink and blue filled diagram insert) from non-SVE WT and $Vipr2^{-/-}$ mice. This was abolished and significantly reduced in post-SVE tissue from WT and $Vipr2^{-/-}$ SCN, respectively (percent changes in amplitude during gabazine treatment: WT non-SVE 20.0 ± 5.6% vs. WT SVE −19.2 ± 13.5%, $p = 0.024$; $Vipr2^{-/-}$ non-SVE 72.1 ± 7.2% vs. $Vipr2^{-/-}$ SVE 31.9 ± 16.2%, $p = 0.025$. All ANOVA with planned comparisons). WT ($mPer2^{luc}$): non-SVE, $n = 7$ and SVE $n = 8$ and $Vipr2^{-/-}$ ($Vipr2^{-/-}$, $mPer2^{luc}$): $n = 7$ each, non-SVE and post-SVE. Abnormal increases in amplitude of non-SVE $Vipr2^{-/-}$ SCN in response to GABA$_A$ receptor blockade are reduced to WT-like responses in post-SVE $Vipr2^{-/-}$ SCN. Gray shaded boxes in **c** represent the interquartile distance between the upper and lower quartile with the median plotted as a horizontal line within the box. Whiskers depict the lower quartile − 1.5 × interquartile distance and upper quartile + 1.5 × interquartile distance. Individual data points are overlaid. *$p = 0.05$. See also Figs. S3 and S4. D and V in diagram inserts label dorsal SCN and ventral SCN, respectively.

VIP–VPAC$_2$ receptor signaling will require substantive further work. Additionally, experimental and simulation research indicate that GABA neurotransmission also couples the dSCN and vSCN subregions, potentially both via positive and repulsive coupling[12,47,77]. In this regard, it is tempting to speculate that exercise stabilizes rhythms by restoring the balance in neural activity between the dSCN and vSCN of $Vipr2^{-/-}$ mice.

A direct mostly excitatory projection from the eye communicates light entrainment information to the SCN[78], but neuropeptide signaling-deficient mice show abnormal synchronization to light. Indeed, the SCN of $Vipr2^{-/-}$[20] and $Vip^{-/-}$ [79] animals are unable to temporally control or "gate" their molecular responses to the activation of this excitatory light input pathway, which likely underpins their aberrant entrainment to LD cycles. However, multiple redundant neural pathways originating in the hypothalamus, thalamus, and brainstem convey arousal-related information to the SCN[80–83], and in contrast to the excitatory glutamatergic light input pathway, arousal efferents utilize inhibitory neurochemicals, including serotonin and neuropeptide Y[64,84]. Exercising in a running wheel suppresses SCN neuronal activity in vivo[85–88] suggesting that recurrent inhibitory input into the $Vipr2^{-/-}$ SCN arising from timed wheel-running acts as a powerful synchronizer, paradoxically enhancing clock cell synchrony to reorganize timekeeping in these mice. Thus, for these neuropeptide signaling-deficient mice, arousal, not light, is the more effective Zeitgeber.

In our investigations, the number of $Vipr2^{-/-}$ mice that subsequently exhibited ~24 h behavioral rhythms was determined by the parameters of timed wheel-running (either the number of hours per day that the wheel was available for exercise (1–6 h) or the number of consecutive days of 6 h/day SVE (8–22 days)). Thus, SVE can be described as "dose-dependently" altering circadian rhythms in neuropeptide signaling-deficient animals. As WT mice adjust much more gradually to SVE, this indicates increased plasticity in the circadian timing system of $Vipr2^{-/-}$ mice. Indeed, $Vipr2^{-/-}$ animals can quickly adjust to advances

or delays in the timing of the SVE regimen. Further, although $Vipr2^{-/-}$ mice rapidly adapt to a 25 h regimen of timed wheel-running, most of these animals subsequently express ~24 h behavioral rhythms on termination of this cycle. Therefore, while these animals have a malleable circadian timing system, their default periodicity appears to be in the range of 22–24 h.

Recent studies have highlighted and differentiated the rhythm generation and timekeeping properties of VIP and VPAC$_2$ receptor expressing cells and subregions in the SCN[89,90]. Our findings build and advance on these as we observe that, following 3 weeks of SVE, mice completely deficient in VIP–VPAC$_2$ receptor signaling ($Vip^{-/-}Vipr2^{-/-}$ animals) can sustain ~24 h behavioral rhythms. Through our investigations, we find that timed wheel-running suppresses $Vipr2^{-/-}$ dSCN neuronal activity through a GABA-independent mechanism. This accompanies SVE-mediated improvements in $Vipr2^{-/-}$ SCN clock cell synchrony (Figs. 3 and 6d–f) with the promotion of stable, near-24 h rhythms in behavior in these mice. Our observations that the cell-coupling opposing actions of GABA in the $Vipr2^{-/-}$ SCN are much reduced by exercise, whereas the inhibitory action of this neurotransmitter is not greatly decreased indicate differential actions of this Zeitgeber on electrical and molecular activities of SCN neurons. It remains to be determined whether and how GABA receptor expression and GABA synthesis in the SCN is influenced by SVE and whether this is critical for the rhythm-promoting effects of timed wheel-running in neuropeptide signaling-deficient mice. Similarly, it is unclear whether other non-SCN circadian oscillators such as those engaged by timed daily food availability are also activated by SVE[91].

Our observation that some mice completely deficient in VIP–VPAC$_2$ signaling ($Vip^{-/-}Vipr2^{-/-}$ animals) spontaneously express short period (~22 h) wheel-running rhythms that can be lengthened by the 24 h Zeitgeber of SVE suggests that other non-VIP-non-VPAC$_2$ receptor signaling mechanisms compensate to enable a modicum of SCN function. It is estimated that VIP and VPAC$_2$ cells constitute 10 and 35%, respectively, of all SCN neurons[92,93]. An implication here then is that an SCN completely "blind" to this neuropeptide signal still retains sufficient plasticity to organize behavioral rhythmicity and respond to an arousal Zeitgeber. Although we do not set out to specifically identify non-GABA signals, potential candidates for residual rhythmicity include SCN cells expressing gastrin-releasing peptide, AVP, or neuromedin S[36,90,94].

Collectively, our findings indicate that, while coherent SCN function ordinarily requires VIP–VPAC$_2$ receptor communication to negate the suppressive action of GABA on molecular and neuronal activity, coherent accelerated behavioral rhythms can be sustained even in the complete absence of VIP and VPAC$_2$ receptor expression. Since evidence indicates that SCN molecular components are subject to differential transcription/posttranslation[95,96], further studies are necessary to scrutinize the actions of exercise on individual clock genes and their proteins. For neuropeptide signaling-deficient mice, synchronization to the LD cycle is abnormal, but they can closely align their circadian rhythms to timed physical exercise. This arousal-related cue differentially alters GABA's suppressive action on neural and molecular activity, improving cellular synchrony and rhythmicity, and promotes 24 h rhythms in behavior. Surprisingly, these actions of timed running-wheel activity in the VPAC$_2$ receptor-deficient mouse are accompanied by a reduction in dSCN spontaneous neural activity, indicating longer-term remodeling of other non-GABAergic mechanisms. The results of this study raise the possibility that regular physical exercise is a suitable stimulus to improve 24 h neural and behavioral rhythms and activity in aged animals, including humans, in which VIP signaling is reduced in the SCN[97–99]. Further, since elderly people can be physically incapacitated and unable to exercise, our findings raise the possibility that

drugs acting to reduce GABA signaling in the SCN may be useful for ameliorating age-related decline in circadian rhythmicity.

## Methods

**Animals**. For experiments included in this study, we utilized adult male mice (>8 weeks) both with and without different reporters of core circadian clock gene/protein expression from the following genotypes: C57BL/6 (WT; Harlan, Blackthorn, UK), $Vipr2^{-/-}$[7] (originally derived from ref. [13]), $mPer2^{luc}$[50] (originally derived from ref. [100]), $Vipr2^{-/-},mPer2^{luc}$[50], $mPer1::$d2eGFP[33] (originally derived from ref. [101]), $Vipr2^{-/-},mPer1::$d2eGFP[33], $Vip^{-/-}$[19], and $Vip^{-/-}Vipr2^{-/-}$ (bred in house at the University of Manchester by crossing the $Vip^{-/-}$ and $Vipr2^{-/-}$ strains). Specific animal numbers used for different experiments are described elsewhere in the "Methods" section, figure legends, and main text. All mice were bred in their respective University of Manchester breeding colonies and maintained at 20–22 °C and humidity of ~40%, with ad libitum access to food and water prior to and throughout experiments. Breeding rooms were maintained on a 12-h:12-h LD (LD$_{12:12}$) cycle. All experiments were performed in accordance with the UK Animals (Scientific Procedures) Act of 1986 using procedures approved by The University of Manchester Review Ethics Panel. Initial breeding stocks of $mPer1::$d2eGFP and $mPer2^{luc}$ mice were kind gifts of D. McMahon and J.S. Takahashi respectively. Initial breeding stocks of $Vipr2^{-/-}$ and $Vip^{-/-}$ animals were kind gifts of A. Harmar and J. Waschek, respectively.

**Behavioral assessment and behavioral paradigms**. During behavioral experiments, mice were singly housed in running wheel-equipped cages with either a contact drinkometer or precision balance to monitor drinking activity. Wheel-running and drinking activities were recorded using either the Chronobiology Kit (Stanford Software Systems, Santa Cruz, CA., USA) or PhenoMaster (TSE Systems, Bad Homburg, Germany) software.

To assess behavioral responses to SVE, mice were maintained for a minimum of 10 days under a LD$_{12:12}$ before transfer to DD for the remainder of the experiment. In DD, mice were initially allowed to free-run for at least 14 days (pre-SVE) followed by either 8 or 19–22 days of SVE (wheel-running restricted to 6 h per day on a repeating 24 h schedule by locking of the running wheel). Wheel-locking/unlocking was performed either manually (see ref. [21] for details) or using an automated system (PhenoMaster with Enable/Disable function, TSE Systems). One cohort of $Vipr2^{-/-}$ mice ($n = 8$) was placed on an SVE protocol, which allowed only 1 h per day of wheel exercise for 21 days (SVE$_{1h}$), and one cohort ($n = 28$) was presented with the opportunity to exercise in the home-cage running wheel for 6 h every 25 h (SVE$_{25h}$), rather than the standard 24 h schedule. For another cohort of $Vipr2^{-/-}$ mice ($n = 18$), we assessed responses to phase-shifts of the standard 6 h/day SVE by advancing and delaying the timing of the opportunity to exercise by 8 h. These mice we allowed to fully entrain to the timing of each SVE schedule for a minimum of 16 days before any shift in the phase of SVE. Drinking activity was taken as a measure of general activity and used to assess behavior during SVE when the running wheel was locked. Following SVE, mice either remained in DD but running-wheel activity was again available ad libitum or were taken directly from SVE for other investigations. Mice were allowed to free-run for a minimum of 14 days for post-SVE behavioral assessment. Mice used as non-SVE controls for all experiments (including long-term DD for $Vipr2^{-/-}$ and $Vip^{-/-}Vipr2^{-/-}$ mice) were singly housed, as described above, for at least 7 days of LD$_{12:12}$, then maintained in DD for durations similar to experimental mice that underwent SVE but with ad libitum access to the running wheel.

**Behavioral analysis**. For behavioral experiments where mice were allowed to free-run both before and after SVE, behavioral parameters were compared between pre- and post-scheduled exercise to assess the effects of SVE on running-wheel locomotor rhythms. The percentage of mice expressing identifiable circadian rhythms in wheel-running was assessed; mice were classified as either rhythmic or arrhythmic (expressing multiple, low power periodic components) pre- and post-SVE based on actograms and periodogram analysis of wheel-running activity (using previously defined criteria[21]). For rhythmic individuals, period and rhythm strength (% variance) were calculated for 14-day epochs of pre- and post-SVE behavior using $\chi^2$-periodogram analysis in Analyze9 (Stanford Software Systems) and Clocklab (Actimetrics, Evanston, IL, USA) software. All period data were also verified by manual assessment of actograms by two experienced experimenters blind to experimental conditions. Drinking activity was monitored throughout the experiment with period and phase assessed using eye-fit regression lines through the onsets of activity. The effects of SVE on locomotor activity rhythms were further quantified by assessing the percentage of pre- and post-SVE that were rhythmic with a period of ~24 h. "~24 h" was defined with reference to the behavior of WT mice used in this study; any period value falling within the range of the mean period of WT mice in pre- and post-SVE ± 2× the mean standard deviation of period for WT pre- and post-SVE. This defined a range of 23.33–24.34 h. To assess the phase of drinking activity onset under SVE, relative to the time of wheel release (and hence wheel-running activity onset), average waveforms were constructed for both wheel-running and drinking behavior during SVE. The $Vipr2^{-/-}$ population assessed for behavioral responses to 3 weeks of SVE ($n = 39$) comprised of mice on the C57BL/6 (non-reporter) background ($n = 18$), as well as

$mPer2^{luc}$ ($n = 15$) and $mPer1$::d2eGFP ($n = 6$). $Vipr2^{-/-}$ mice on different reporter/non-reporter backgrounds responded similarly to SVE (Fig. S3a, b), so were collapsed into one group. The WT population assessed for behavioral responses to 3 weeks of SVE ($n = 28$) comprised of mice on the C57BL/6 (non-reporter) background ($n = 15$), as well as $mPer2^{luc}$ mice ($n = 13$). Both lines of WT mice used are maintained on a C57BL/6 background and responded similarly to SVE, so were collapsed into one group.

**Immunohistochemistry.** Group-housed mice under $LD_{12:12}$ were culled during the light phase for anti-VIP (1:2000; Enzo Life Sciences, Exeter, UK) and anti-VPAC$_2$ (1:5000; Abcam, Cambridge, UK) nickel di-aminobenzine immunohistochemistry. Brains were processed using standard techniques[33].

**Culture and slice preparation.** Mice used for MEA recordings, confocal imaging, and assessment of PER2-driven luciferase expression under GABA$_A$ receptor blockade were not maintained through a post-SVE epoch but culled immediately after SVE. Mice used for assessment of PER2-driven luciferase expression in the SCN without GABA$_A$ receptor blockade were culled during post-SVE behavior, 10–14 days after the end of SVE. All mice were euthanized by cervical dislocation following isoflurane anesthesia (Baxter Healthcare Ltd, Norfolk, UK) and enucleated in darkness with the aid of night-vision goggles.

For luminescence and fluorescence experiments, mid-SCN-containing brain slices were micro-dissected and cultured as 250-μm-thick coronal slices. Micro-dissected explants were isolated and prepared for culture as previously described[33,80,102,103] using standard plastic-based culture dishes (Corning, UK) for photomultiplier tube (PMT) and lumicycle luminometry and glass coverslip-based dishes (Fluorodish, World Precision Instruments Ltd., Stevenage, UK) for luminescence and fluorescence imaging. We have previously reported phase-resetting of $Vipr2^{-/-}$ SCN tissue during explant preparation[50] but here find no impact of this on the ability of in vitro SCN explants from non-SVE and post-SVE mice to report on SCN cellular synchrony. Where the dorsal and ventral subregions of the SCN were cultured separately, mid-SCN slices were bisected manually using a scalpel with reference to a mouse brain atlas[104] and visual anatomical cues in the slices (see Fig. S4a for approximate location of dorsal/ventral division).

For MEA recordings, mice were culled at the following times: non-SVE mice (of WT and $Vipr2^{-/-}$ backgrounds) were culled during the subjective day at CT9–11 (with CT12 defined as the time of activity onset; non-SVE control mice with no overt behavioral rhythms were culled at random times); post-scheduled exercise $Vipr2^{-/-}$ mice were culled 1–3 h prior to scheduled wheel availability, a time that also corresponded to 1–3 h prior to activity onset as these mice maintain behavioral activity rhythms in phase with the opportunity to exercise. As WT mice stably entrain ~8 h advanced of the opportunity to exercise, post-scheduled exercise WT mice were culled at two different times to control for the time of exercising in the running wheel and endogenous behavioral (drinking) onset, separately. These two WT groups are referred to as SVE(1) and SVE(2), respectively. SVE(1) WT mice were culled 1–3 h prior to the time of wheel availability and SVE(2) WT mice were culled 1–3 h prior to endogenous behavioral onset (Fig. S5). Both groups of WT SVE mice received the same SVE paradigm, only differing in the phase of cull time and subsequent recording phase on the MEA. Recordings were initiated ~2–3 h after cull. Coronal brain slices containing the mid-SCN were prepared as previously described[80,105] and maintained in the MEA recording chamber in which they were continuously perfused (1.8 ml/min MEA) with well-gassed (95% O$_2$; 5% CO$_2$) recording artificial cerebrospinal fluid of standard composition (in mM: NaCl 127; KCl 1.8; KH$_2$PO$_4$ 1.2; CaCl$_2$ 2.4; MgSO$_4$ 1.3; NaHCO$_3$ 26; glucose 15; Phenol Red 0.005 mg/l; pH 7.4, measured osmolarity 300–310 mOsmol/kg)[80,105].

**Luminometry and luminescence imaging.** For basic luminometry assessment of control and post-SVE SCN, cultures were maintained at 37 °C in light-tight incubators (Galaxy R+, RS Biotech, Irvine, Scotland) and total PER2::LUC bioluminescence emission recorded for 7 days using PMT assemblies (H8259/R7518P; Hamamatsu, Welwyn Garden City, UK). Emitted photon counts were integrated for 299 s every 300 s and raw bioluminescence data were baseline-subtracted with a 24-h running mean, then smoothed with a 3-h running average. The longitudinal study design employed here allows sensitive identification of low-amplitude rhythms in individual animals, such as those of $Vipr2^{-/-}$ mice. Discontinuous sampling methods, which assess population-level trends across a number of individuals, can fail to detect significant variation when individuals are not synchronized to one another or peak–trough amplitude is low[7,13,20].

For luminometry assessment of SCN rhythms before and during GABA$_A$ receptor blockade in control and post-SVE tissue, cultures were maintained at 37 °C in a LumiCycle system (Actimetrics) housed within a light-tight incubator (Galaxy R+, RS Biotech). Total PER2::LUC bioluminescence emission was recorded for 8 days. On the fifth day cultures were treated with 100 μM gabazine by addition of 1 μl of 100 mM gabazine directly to the existing culture media, without media change. As gabazine was dissolved in 100% dimethyl sulfoxide (DMSO), control cultures were treated with 1 μl DMSO to give a final concentration of 0.001% DMSO in both gabazine-treated and control cultures. Emitted photon

counts were integrated for 60 s every 600 s and bioluminescence profiles smoothed and de-trended as described for PMT data above.

**Fluorescence imaging.** $mPer1$::d2eGFP fluorescence was imaged with a C1 confocal system running on a TE2000 inverted microscope (Nikon, Kingston, UK) using a ×10 0.3NA Plan Fluotar objective (Nikon). A 488-nm laser line was used for excitation and emitted fluorescence detected using a 515/30 nm band pass filter cube. One 16-image "Z" stack was acquired every hour for the duration of recording, using ×3 Kalman averaging at ×1 confocal zoom with an open pinhole and a pixel dwell of 12.96 μs. Each stack covered a total depth of 60.8 μm and images were recorded at a resolution of 512 × 512 pixels. Z stacks were collapsed to an average projection using ImageJ and fluorescence expression profiles of 30 individual cells were selected at random. Raw fluorescence data were corrected for variations in background brightness by subtracting the optical density value of a standardized, non-GFP-expressing, non-SCN region from each data value before corrected data were smoothed using a 3-h running mean.

**Luminescence and fluorescence analysis.** For basic luminometry assessment of rhythms in control and post-SVE explants, rhythmic traces were measured as described previously[50] to extract the period and amplitude of rhythms. For luminometry assessment of SCN rhythms before and during GABA$_A$ receptor blockade in control and post-SVE SCN, peak–trough rhythm amplitude was measured for the last cycle before and the first cycle 24 h following treatment with gabazine or DMSO. These values were used to calculate the post-gabazine/pre-gabazine fold change in rhythm amplitude. Changes in rhythm amplitude in response to control DMSO treatments were small (~0.5-fold) and did not significantly differ between groups.

Rhythmicity of fluorescence traces from putative single cells was assessed by two experienced, independent researchers blind to conditions and genotype. Rhythmic traces were measured as described previously[34,50] to extract the percentage of rhythmic cells and synchrony between cells within slices, as well as period variability between cells within slices (defined as standard deviation of mean period for all rhythmic cells within a slice). Since previous research showed that $Per1$-driven eGFP expression is very low in VIP–VPAC$_2$ signaling-deficient mice[33], for display purposes, fluorescence data were normalized to an arbitrary maximum to aid in visual assessment of rhythm profiles in different slices. To investigate dorsal–ventral subregional differences in SCN circadian function and responses to SVE, analyzed cells providing fluorescence data were classified as either dorsal or ventral, based on anatomical characteristics of each slice, with reference to the mouse brain atlas[104] and recently published emergent clusters within the SCN[47,106–109].

**MEA recording of action potential discharge.** MEA data were recorded at 33 °C using a dual-MEA2100-HS2x60 system (Multi-Channel Systems (MCS) GmbH, Reutlingen, Germany) with 60pMEA100/30iR-Ti-gr MEAs. Raw MEA data were collected at 25 kHz in MC_Rack (MCS) and high-pass filtered offline using a 300 Hz Butterworth second-order filter. Events (action potentials) were discriminated using a threshold set at −17 μV. This threshold was confirmed as appropriate for every recording by assessing system noise at the end of recording sessions following a treatment with 1 μM tetrodotoxin (Tocris Bioscience, Bristol, UK). Time series of extracted events were smoothed using a 10-point boxcar filter in Neuroexplorer (Nex Technologies, Madison, Alabama) and mean action potential firing of regions of interest in the time series (initial baseline firing; pre-treatment and response firing around gabazine treatments) assessed using Spike2 (Cambridge Electronic Designs, Cambridge, UK). Responses to gabazine treatments were considered significant if the mean post-treatment response was greater than mean pre-treatment firing plus 2 standard deviations of mean pre-treatment firing. Once responses were determined to be significant using this conservative threshold, absolute pre-post-treatment responses to gabazine were used for analysis. Heat-maps were created using Excel and Origin Pro (OriginLab Corp., Northampton, MA).

**Statistics and reproducibility.** Statistically significant differences in continuous measures of bioluminescence, fluorescence, and behavioral data were determined, as appropriate, using paired $t$ tests (Microsoft Excel), or two-way analysis of variance (ANOVA), with a priori pairwise comparisons (SYSTAT 10, SPSS, Chicago, IL). Firing rate data from MEA experiments were initially assessed for genotype, SCN subregion, and exercise condition by three-way ANOVA (JMP ver 14, SAS Institute Inc., Cary, NC). Subsequently genotype and exercise condition differences within the dorsal subregion or ventral subregion were assessed by one-way ANOVA and Tukey honest significance test post hoc comparisons (Kaleidagraph ver 4.5.4, Synergy Software, Reading, PA). In addition, the synchrony of individual cells within explants was assessed using Rayleigh vector plots performed with custom software designed in house by Prof. T. Brown, as well as the El Temps software (Dr. A. Diez-Noguera, Barcelona, Spain). Significant changes in the percentage of behaviorally rhythmic mice between pre- and post-SVE were assessed using McNemar's test (Graphpad online calculator: www.graphpad.com/quickcalcs/). Box plots with overlaid dot plots were made using Kaleidagraph ver. 4.5.4 (Synergy Software, Reading, PA). Specific details of tests used, outcomes,

sample sizes, summary values, and dispersion can be found in the figure legends and main text. For all statistical tests, $p < 0.05$ was required for significance.

**Reporting summary**. Further information on research design is available in the Nature Research Reporting Summary linked to this article.

## Data availability
Datasets are available by request to the corresponding author.

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

## Acknowledgements

We thank Carla Santos da Silva and Cheryl Petit as well as Dr. Sandrine Dupre and Dr. Andrea Power for technical assistance. This work was funded by project grants from the Biotechnology and Biological Sciences Research Council (BBSRC; BB/J003441/1 and BB/M02329X/1 to H.D.P. and J.G. at the University of Manchester and BB/R019223/1 to H.D.P. and J.G. at the University of Bristol and the University of Manchester), a PhD studentship to S.W. from the University of Manchester Neuroscience Research Institute, a project grant from the Wellcome Trust (MA086352) to H.D.P. at the University of Manchester, and funding from the Human Frontiers of Science Programme (RGP0030/2015) and Wellcome Trust (107851/Z/15/Z) to A.S.I.L. at the University of Manchester.

## Author contributions

A.T.L.H., R.E.S., and H.D.P. designed the studies. A.T.L.H., R.E.S., C.G., and R.C.N. performed behavioral experiments. A.T.L.H. and R.E.S. analyzed behavioral experiments. A.T.L.H., B.B.-O., S.W., and M.D.C.B. designed, recorded, and analyzed the multi-electrode array studies. A.T.L.H., C.G., and R.C.N. made and analyzed bioluminescence and fluorescence recordings. A.T.L.H., H.D.P., J.G., and A.S.I.L. wrote the manuscript.

## Competing interests

The authors declare no competing interests.
