## [Peer Review File · Communications Biology]

Reviewers' comments:

Reviewer #1 (Remarks to the Author):

In this study, Hughes et al. demonstrated that daily scheduled voluntary exercise (SVE) promoted persistent circadian behavioral rhythms in mice deficient Vip signaling in the SCN. Then they measured Per1-GFP rhythms at the single cell levels in the SCN, and found that SVE increased synchrony and rhythmic cells in the Vipr2^{-/-} SCN cells. They also examined the effect of SVE for neuronal firing in the SCN using MEA. SVE differently modulate firing rate in the SCN between WT and Vipr2^{-/-} mice. Pharmacological inhibition of GABA_A signaling in cultured SCN revealed that SVE suppressed the GABAergic contribution to spontaneous firing in the dorsal SCN. However, SVR did not have significant effects for inhibitory effects of GABA on firing in the ventral SCN of Vipr2^{-/-} mice. Gabazine evoked increases in the amplitude of PER2::LUC rhythms of WT and Vipr2^{-/-} SCN were attenuated by SVE.

This is an interesting and informative paper that provides role of an exercise (non-photic stimulation) and Vip signaling in the SCN for circadian rhythms. The experiments were adequately performed, but there are several points for the authors to consider.

1. The authors used Per1-GFP and PER2::LUC mice to see the effects of SVE to molecular rhythms in the SCN. Since Per1 and Per2 expression rhythms do not always show the same special-temporal patterns in the SCN, and transcription (Per1-GFP) and protein (PER2::LUC) rhythms also do not always show the same patterns (due to post transcriptional translational modification), they had better to use one reporter mice in the present study. Indeed, SVE did not have any effects for Per1-GFP rhythms in the WT SCN, but amplitude of PER2::LUC rhythms in the SCN was attenuated by SVE. Is this due to different genes or reporters (promotor activity v.s. protein)? If the authors would like to use both reporter mice, they had better to clearly consider these issues.
2. The authors described the results of Vip^{-/-}: Vipr2^{-/-} behavioral data in figure 1a, but the detailed description are appeared after Figure 2 in the manuscript (L148). For easy readability, I recommend the authors to move the paragraph after the first paragraph of the result (L118).
3. The authors measured neuronal activity using MEA. The SCN was cultured CT9-11 from non-SVE control, and 1-3h prior to scheduled wheel availability from post-SVE. Since some of WT mice entrained SVE but others did not, circadian phase of firing rhythms might not be the same between non-SVE and post-SVE.
4. The authors found that SVE increased circadian rhythmicity of behavioral rhythms in KO mice. How about the amplitude of behavior? Since the Per1-GFP data showed that SVE increased synchrony of rhythms in the Vipr2^{-/-} SCN, SVE might increase the amplitude of behavior. In addition, because KO mice seemed to entrain to SVE, it is interesting to analyze phase angle between onset of SVE and activity onset after SVE.
5. In this paper, behavioral rhythms are important results. I recommend the authors show further behavioral data in the supplement. Also I recommend they moved Fig1b into the supplement.
6. After SVE, percent mice rhythmic at 24 hours increased in behavioral rhythms in KO mice (Figure 1f). However circadian period in the SCN were not different between SVE and non-SVE group (Figure 4b and d). How can the authors explain the results? The period differences were observed in Per1-GFP reporter?
7. L132: It is better to insert "in WT and Vipr2^{-/-} SCN" at the end of the sentence.
8. The authors used "timed wheel-running" and "scheduled voluntary exercise" in the script. Do they mean the same?

9. The authors showed that SVE have a long term remodeling in the SCN which influence to behavioral rhythms in KO mice. It is interesting to know how long SVE exposure is required to show this remodeling. They used 3 weeks SVE and found behavioral changes in KO mice. Is this change observed in several days in SVE? It is interesting to identify critical length of SVE which needs to obtain the remodeling.

10. Some KO mice did not show 24 rhythmicity after SVE (Figure 1f). Is there any correlation between number of wheel running and behavior in KO mice?

11. L819: The authors mentioned that "In DD, mice were initially allowed to free-run for at least 14 days (pre-SVE) followed by either 8 or 19-22 days of SVE (wheel-running restricted to 6h per day by locking of the running wheel.)" What is the reason why they used 8 days of SVE?

12. Figure2b: Please clearly show how the fluorescence data were normalized. It is better to show cell location in the SCN in Fig2a that was shown in Fig2b. For easy understanding, it is better to insert Rayleigh plots and movies. Please inset scale bar in Fig2a.

13. Figure 3: It is better to insert raw data of firing with Gabazine application. Please indicate how the heatmaps were made.

14. Figure 4: The authors showed PER2::LUC rhythms in dorsal or ventral –only the SCN. Since they used whole SCN and measured firing in Figure 3, it is better to show circadian rhythms in intact SCN, and show amplitude and period. Otherwise, it is difficult to compare firing and PER2::LUC data. Physical cut between the dorsal and ventral SCN shows different characteristics of circadian rhythms in some case as compared with whole SCN (Albus et al., 2005 Current Biol). Also it is better to show PER2::LUC rhythms before and after Gabazine application similar to Figure 4a and c.

15. In the present study, the authors explained that behavioral change by SVE was due to functional changes of GABA function in the SCN. However they have not shown direct evidence that GABA in the SCN was critical for the modulation of behavior by SVE. Additionally, because several oscillator models have been suggested (such as FEO and MASCO), SVE might have an effect for peripheral oscillators that regulate behavior. Indeed, discrepancy between the SCN phenotype and behavior by SVE were observed (reduced amplitude in PER2::LUC and no-change of behavior in WT; no change of PER2::LUC period and 24h rhythms in KO mice after SVE), It is important to identify the roles of peripheral oscillates influenced by SVE.

16. The authors had better to refer this paper (Reebs and Mrosovsky JBR 1989) in the introduction. Reebs demonstrated the effects of wheel running for circadian behavior.

Reviewer #2 (Remarks to the Author):

The authors present a strong and thorough investigation into how circadian rhythms in mice with compromised VIP signalling can be improved through scheduled exercise. This work build on their previous work in this area (Power et al. 2010). In the present paper they replicate the behavioural findings from the previous work, and extend it with a number of in vitro investigations looking at both electrical activity and clock gene expression as assessed with a luciferase reporter. The quality of the work is excellent and the data support their conclusions. I have some suggestions for improving the presentation, but overall the work is solid and of high quality.

Comments:

I was initially confused by the use of "~24h", since this is being used in a different manner than

"circadian rhythm" (which by convention in the field is also ~ 24 h). I think given the style of this journal that the definition needs to be repeated in the results section rather than just in the method. The text needs to be sharpened around this. I would recommend highlighting figure 1e first, and then include an analysis showing that when a rhythm is detected in the pre-SVE days, that it is significantly different than that of WT mice. The rationale for the ~ 24 h definition isn't clear. Is this to match them to WT periods, or could it be that this is actually matching the periods of the SVE schedule. Put another way, what would happen if they put these mice on a non-24h T cycle of SVE? Would the exercise increase the strength of the network to yield a WT-like period, or would the period match more closely the T-cycle period?

The ephys work would benefit from highlighting in the results section the phase. Some expansion of the rationale for this phase would also be warranted, as CT13 is generally quite late for SCN ephys, with the SCN showing more activity in the mid-subjective day. An investigation of changes of electrical activity across the day would have been interesting, and might be something to consider in follow-up studies. Some of the differences observed could be due to simply sampling non-comparable phases in the various experimental groups, rather than being due to more profound and long term changes in GABA signalling.

The data in figure 2 are really exciting as they get around the phase sampling issue mentioned above with respect to the ephys work. It would be helpful if the phase for the photos in panel A was indicated on the traces in panel B.

Figure 4 is quite nice as well, but thinking back to the analysis in figure 2, one is left wondering if the changes in tissue level rhythms are due to changes in the individual cellular rhythms, or due to changes in phase coherence of the population. Do the authors have data to draw upon, such as in figure 2, to address this?

How representative is the trace in the upper left of figure 3d? This seems a lot noisier and more variable than the other traces. I note that the bar in 3c for this trace has more variability than the other bars. This particular trace looks like it would be much higher than ~ 9 Hz. A horizontal line representing the discrimination threshold in each trace might help the reader.

Figure 3g-j represent the experiment with gabazine, although this isn't clear in the figure. The time of application could be indicated in panel 3h, and the word gabazine could appear somewhere in these panels to help the reader. The same can be suggested for figure 4e.

In the conclusion (and end of the intro) the authors argue that exercise could be used to restore rhythmicity in people with weak clocks such as the elderly. Is there evidence that the VIP/VPAC2 KO mice model similar problems that are observed in the elderly? The authors might also want to consider the interplay between activity and rhythmicity in the elderly. In many cases the lack of activity might be the cause of the weak rhythms. Also those with aging related problems in their rhythms might not be able to exercise. In that sense the work described here investigating the neural mechanisms are critical, as it might help identify interventions that could yield the same effect without requiring exercise itself.

Reviewer #3 (Remarks to the Author):

This group has previously reported that *vip*^{-/-} and *vipr2*^{-/-} mice exhibit weak, short tau rhythms in DD that can be entrained to 24h scheduled wheel access, and that show longer taus (aftereffects) in DD following entrainment. In these KO models, scheduled wheel running appears to be a more potent Zeitgeber than LD cycles. The present study confirms these results and describes cellular correlates in the SCN pacemaker, using single cell clock gene imaging and electrophysiology. The results suggest that nonphotic entrainment in the KO mice is facilitated by down-regulation of inhibitory GABA

signaling associated with scheduled activity.

results

1. line 109 - The authors state that "mice deficient in VPAC2 (*Vipr2*^{-/-}) or VIP (*Vip*^{-/-}) show apparent daily rhythms under the light-dark cycle but cannot express endogenous ~24h rhythms in behavior when placed in constant dark." The wording is potentially confusing, because the supporting figures clearly show that the KO mice express significant rhythmicity, which could be defined as in the 'circadian range' (e.g., WT mice can entrain to 22h T cycles, which places 22h in the circadian range). This is a minor point, but I would substitute more descriptive wording, and say instead that the KO mice express endogenous rhythms with a periodicity of 22 +xx h, which is significantly shorter than the circadian periodicity of 23 + xx h in WT (and differences in amplitude, precision, etc., whatever else might obtain).

2. I would also avoid saying that the KOs "cannot" express ~24h rhythms, because you then immediately show that they can (i.e., following the wheel access schedules). The KO mice show an aftereffect on tau of entrainment to the 24h activity schedule that is not evident following exposure to 24h LD cycles.

3. line 117 - The WT mice did not show an effect of the activity schedule on tau, but the schedule wasn't in place long enough for the WT mice to entrain, so one would not expect to see an aftereffect similar to that evident in the KO mice (aftereffects being a response to entrainment). When the activity schedule is maintained until the WT do entrain, do they show evidence of aftereffects?

4. There were differences between WT and *vipr2*^{-/-} mice in dorsal SCN firing rates. Meaningful comparison is predicated on sampling at the same circadian phase in the two groups. Many of the KO mice did exhibit significant rhythmicity prior to scheduled activity, and thus a CT could have been defined by reference to the regression lines, such as illustrated in Figure 1. In cases where the recordings were made from KO mice with significant rhythmicity, have you tried grouping the recordings according to CT, for comparison with the WT mice, which were sampled at CT9-11?

5. Line 236 - "Coincident with the post-SVE neurophysiological changes in the WT SCN, we recorded a significant reduction in the amplitude of PER2::LUC oscillations following SVE in both doSCN and voSCN from these neuropeptide-competent mice". Were these recordings from WT mice that entrained to SVE, and if not, do you think amplitude would be increased in WT mice culled after entrainment was achieved?

6. The results support the authors' conclusion that nonphotic entrainment in the KO mice is facilitated by down-regulation of inhibitory GABA signaling associated with scheduled activity. This is a puzzling effect, given that the manipulated variable is wheel running; if scheduled running downregulates GABA, why does downregulation not occur when the mice have free access to the wheel?

We thank the reviewers for their expert comments and note that they are collectively very positive. We apologise for the delay in getting the revised manuscript submitted, but unfortunately due Covid-19 restrictions, key data necessary for this were on hard-drives that were locked in an office that we were not permitted to enter until the end of July. We have now included new data to address some of points raised and revised the manuscript accordingly.

Reviewers' comments:

Reviewer #1 (Remarks to the Author):

In this study, Hughes et al. demonstrated that daily scheduled voluntary exercise (SVE) promoted persistent circadian behavioral rhythms in mice deficient Vip signaling in the SCN. Then they measured Per1-GFP rhythms at the single cell levels in the SCN and found that SVE increased synchrony and rhythmic cells in the Vip2^{-/-} SCN cells. They also examined the effect of SVE for neuronal firing in the SCN using MEA. SVE differently modulate firing rate in the SCN between WT and Vip2^{-/-} mice. Pharmacological inhibition of GABA signaling in cultured SCN revealed that SVE suppressed the GABAergic contribution to spontaneous firing in the dorsal SCN. However, SVR did not have significant effects for inhibitory effects of GABA on firing in the ventral SCN of Vip2^{-/-} mice. Gabazine evoked increases in the amplitude of PER2::LUC rhythms of WT and Vip2^{-/-} SCN were attenuated by SVE. This is an interesting and informative paper that provides role of an exercise (non-photic stimulation) and Vip signaling in the SCN for circadian rhythms. The experiments were adequately performed, but there are several points for the authors to consider.

1. The authors used Per1-GFP and PER2::LUC mice to see the effects of SVE to molecular rhythms in the SCN. Since Per1 and Per2 expression rhythms do not always show the same special-temporal patterns in the SCN, and transcription (Per1-GFP) and protein (PER2::LUC) rhythms also do not always show the same patterns (due to post transcriptional translational modification), they had better to use one reporter mice in the present study. Indeed, SVE did not have any effects for Per1-GFP rhythms in the WT SCN, but amplitude of PER2::LUC rhythms in the SCN was attenuated by SVE. Is this due to different genes or reporters (promotor activity v.s. protein)? If the authors would like to use both reporter mice, they had better to clearly consider these issues.

The reviewer raises a good point. To mitigate this, we already include a clear rationale for the use of both reporter strains in the Results text (lines 191-197 and 302-307). We used different lines as the cellular resolution is better with the Per1::eGFP line, while the duration of recording is elongated with the more passive bioluminescence monitoring system. One way to have addressed this would have been to use mouse lines crossed to the Per1::Luc background. Unfortunately, we did not have that line available to us during these investigations. The reviewer is correct that we found no differences in the parameters of Per1-GFP expression in WT SCN as a result of SVE but did find differences in the amplitude of PER2::LUC. We note, however, that the parameters assessed for Per1-GFP and PER2::LUC experiments presented here differ (due to the specific aims of each

experiment) and it is equally likely that the differences observed are unique to the parameters assessed.

Importantly, in this study, we show that the behavioral effects of SVE were similar between non-reporter mice and their corresponding reporter-expressing strains. Thus, the inclusion of reporter constructs per se does not affect the behavioral outcome of SVE (Fig. S3).

2. The authors described the results of *Vip*^{-/-}: *Vipr2*^{-/-} behavioral data in figure 1a, but the detailed description are appeared after Figure 2 in the manuscript (L148). For easy readability, I recommend the authors to move the paragraph after the first paragraph of the result (L118).

We thank the reviewer for this suggestion. We certainly contemplated this prior to our original submission, but felt that the reporting of the *Vip*^{-/-}:*Vipr2*^{-/-} experiments flows better when positioned in its current location in the Results. We contend that this is the preferred location for this text as it allows us to smoothly introduce the idea that other neurotransmitters potentially contributed to the SCN re-structuring effects of scheduled exercise.

3. The authors measured neuronal activity using MEA. The SCN was cultured CT9-11 from non-SVE control, and 1-3h prior to scheduled wheel availability from post-SVE. Since some of WT mice entrained SVE but others did not, circadian phase of firing rhythms might not be the same between non-SVE and post-SVE.

All WT mice used for MEA recordings were stably entrained to SVE (existing example shown in Fig 1b and an additional example has been added to Fig S1 (Fig. S1f)). The stated collection/recording times were chosen as they represent the same phase relationship between exercise and collection /recording for all genotypes. Further, the phase angle of entrainments observed here for WT and *Vipr2*^{-/-} mice are similar to those that we observed in our previous research on SVE in these mouse genotypes (Power et al., 2010).

4. The authors found that SVE increased circadian rhythmicity of behavioral rhythms in KO mice. How about the amplitude of behavior? Since the *Per1*-GFP data showed that SVE increased synchrony of rhythms in the *Vipr2*^{-/-} SCN, SVE might increase the amplitude of behavior. In addition, because KO mice seemed to entrain to SVE, it is interesting to analyze phase angle between onset of SVE and activity onset after SVE.

These are both good suggestions. We have added a new panel to Fig. 1 (Fig. 1h) and now report data on the amplitude of behavioral rhythms. Information on phase angle of entrainment of behavioral rhythms to SVE is already included in the original submission (Fig. S1e).

5. In this paper, behavioral rhythms are important results. I recommend the authors show further behavioral data in the supplement. Also I recommend they moved Fig1b into the supplement.

We have included additional behavioral actograms, including those mentioned above in response to point 3 (Fig. S1f), actograms from *Vipr2*^{-/-} mice exposed to 8 days of SVE (Fig. 2a-b), for *Vipr2*^{-/-} mice exposed to a 1h/day SVE protocol (Fig. 2c) and for *Vipr2*^{-/-} mice exposed to a 25h SVE Zeitgeber (Fig. 2d-f). We have also added details of behavioral analysis for these to Table 1 as well as adding rhythm strength analyses to

6. After SVE, percent mice rhythmic at 24 hours increased in behavioral rhythms in KO mice (Figure 1f). However circadian period in the SCN were not different between SVE and non-SVE group (Figure 4b and d). How can the authors explain the results? The period differences were observed in Per1-GFP reporter?

The reviewer raises a good point. When surveying studies from different labs, including our own, it is clear that SCN period of molecular clock reporters can differ from that of the behavioral rhythm (e.g. Maywood et al., 2006, *Curr Biol*; Hughes et al., 2008, *J. Neurochem*; Timothy et al, 2018, *Biol Psychiatry*; Mieda et al., *Curr. Biol.* 2016; Loh et al, 2011, *J Biol Rhythms*). In the context of the present work, we believe that the focus should be on SCN cellular synchrony rather than the period as behavioral improvements correlate very well with the increase in SCN cellular synchrony. Moreover, Fig. 4b and d show data for dorsal only (do) and ventral only (vo) micro-dissections of SCN, rather than complete coronal sections through the SCN containing both its dorsal and ventral parts.

7. L132: It is better to insert “in WT and *Vipr2*^{-/-} SCN” at the end of the sentence.

We have done this.

8. The authors used “timed wheel-running” and “scheduled voluntary exercise” in the script. Do they mean the same?

Yes, we mean the same thing—we used different wording to reduce the repetitiveness of the text and include a definition of timed wheel-running as SVE at its first use in the text (line 112).

9. The authors showed that SVE have a long term remodeling in the SCN which influence to behavioral rhythms in KO mice. It is interesting to know how long SVE exposure is required to show this remodeling. They used 3 weeks SVE and found behavioral changes in KO mice. Is this change observed in several days in SVE? It is interesting to identify critical length of SVE which needs to obtain the remodeling.

As noted above, we have now included previously unpublished data on the effects of 8 days of SVE (new Fig. S2b,c, additional information in Table S1) – at a group level, this is not very effective for entrainment to SVE, though some individuals (a reduced proportion compared to 21 days of SVE) did show signs of entrainment and lasting post-SVE effects on behavioral period.

10. Some KO mice did not show 24 rhythmicity after SVE (Figure 1f). Is there any correlation between number of wheel running and behavior in KO mice?

We previously found a significant inverse correlation between wheel-running intensity and latency to entrainment to SVE in WT mice (Power, Hughes et al (2010), but no such relationship was seen for *Vipr2*^{-/-} mice. This is presumably because the vast majority of *Vipr2*^{-/-} individuals rapidly synchronize behavioral rhythms to the opportunity to voluntarily exercise in a running wheel (please see Power, Hughes et al (2010) Fig. S1 for more details). We have very occasionally found mice that do not voluntarily use a homecage running wheel and clearly such individuals cannot be included in these experiments. This, however, is a very rare occurrence and no such mice were excluded from the experiments presented in this manuscript. Moreover, on occasion some WT mice fail to stably entrain

to SVE within an acceptable timeframe (acceptable based on use of experimental resources and economical considerations) and were not used in the present studies.

11. L819: The authors mentioned that “In DD, mice were initially allowed to free-run for at least 14 days (pre-SVE) followed by either 8 or 19-22 days of SVE (wheel-running restricted to 6h per day by locking of the running wheel.)” What is the reason why they used 8 days of SVE?

We have addressed this question above; we had initially intended to show the 8 day SVE data and removed it but unfortunately neglected to remove reference to it. We have now re-integrated this data into the manuscript. These findings indicate that for most mice, more than 8 days of SVE is required to alter the period of the post-SVE behavioral rhythm.

12. Figure2b: Please clearly show how the fluorescence data were normalized. It is better to show cell location in the SCN in Fig2a that was shown in Fig2b. For easy understanding, it is better to insert Rayleigh plots and movies. Please inset scale bar in Fig2a.

We have added new descriptions to the Methods (lines 1200-1203) and included the requested details.

13. Figure 3: It is better to insert raw data of firing with Gabazine application. Please indicate how the heatmaps were made.

Firing rate changes are subtle and very difficult to visualize on raw MEA data. This, traces showing firing-rate changes are a much more effective means to visualize the effects of gabazine application on SCN electrical activity. Details of how heatmaps were made are now included in the methods section (lines 1225-1226).

14. Figure 4: The authors showed PER2::LUC rhythms in dorsal or ventral –only the SCN. Since they used whole SCN and measured firing in Figure 3, it is better to show circadian rhythms in intact SCN, and show amplitude and period. Otherwise, it is difficult to compare firing and PER2::LUC data. Physical cut between the dorsal and ventral SCN shows different characteristics of circadian rhythms in some case as compared with whole SCN (Albus et al., 2005 Current Biol). Also it is better to show PER2::LUC rhythms before and after Gabazine application similar to Figure 4a and c.

We have previously attempted the whole SCN PER2::LUC rhythms experiment suggested here, but unfortunately due to a procedural issue, the data were not usable and, as explained above, we will not be in a position to repeat the experiments for the foreseeable future. We do, however, include a fuller description of the inclusion of dorsal- and ventral –only SCN microdissections to help prevent any misinterpretation of the data (lines 301-302; 1129-1132). Unfortunately, due to difficulties in stabilizing these reduced dorsal-only or ventral-only slices on the MEA platform, it was not possible to make acute electrophysiological recordings from them in the way we did with intact coronal SCN slices.

15. In the present study, the authors explained that behavioral change by SVE was due to functional changes of GABA function in the SCN. However they have not shown direct evidence that GABA in the SCN was critical for the modulation of behavior by SVE. Additionally, because several oscillator models have been suggested (such as FEO and MASCO), SVE might have an effect for peripheral oscillators that regulate behavior. Indeed, discrepancy between the SCN phenotype and behavior by SVE were observed (reduced amplitude in PER2::LUC and no-change of behavior in WT; no change of PER2::LUC period and 24h rhythms in KO mice after SVE), It is important to identify the roles of peripheral oscillators influenced by SVE.

The reviewer has pinpointed several worthy experiments that are the subject of our follow-up grant application on this topic and we have now acknowledged these points in our revised Discussion (lines 416-418; 461-465). To undertake a comprehensive investigation of GABA's critical role here is not trivial and will take considerable time and resources. We agree that there is considerably more to be done to determine how necessary GABAergic transmission is for exercise to exert its actions on the SCN and behavioral rhythms. Such experiments are necessarily complex and time-consuming, involving approaches to eliminate GABA synthesis and GABA receptor expression in the SCN. Since GABA is also present in major neural inputs to the SCN, including the geniculohypothalamic tract (GHT; arising from the visual thalamus), the median raphe, and the retina, additional studies would be required to eliminate GABA synthesis in these inputs. Unfortunately, we are not in a position to initiate these studies as ATNH's institution does not have the facility for behavioral studies, while the animal unit at HDP's current institution is closed for construction work (to make it more Covid-19 secure). Thus, we cannot attempt to conduct new studies to investigate GABA for the foreseeable future. The most obvious approach would be to use Cre mouse lines and viral vectors to reduce GABA expression. One limitation though is that it is now acknowledged that the inclusion of Cre can result in unintended consequences. For example, a widely used Vip-Cre line has reduced VIP expression in the SCN (Cheng et al (2019); Joye et al (2020), both Journal of Biological Rhythms) and, thus, is not the best model for studying exercise and VIP signalling. Therefore, we have acknowledged the limitation of our conclusions in the Discussion (lines 461-465).

The idea of a peripheral signal playing a role is interesting. Data on the Food-Entrainable Oscillator in *Vipr2*^{-/-} mice indicates that the molecular clock is still intact in peripheral tissues of *Vipr2*^{-/-} mice (Sheward et al., 2007 J. Neurosci.). Further, as shown in the Sheward et al. study, the FEO typically rapidly loses its influence on behavioral rhythms when a rodent is returned to ad libitum feeding conditions, whereas here we see a very long lasting effect of 1SVE on behavioral rhythms and SCN. This indicates that different processes underpin the rhythm promoting actions of these stimuli. Similarly for the MASCO, we do not know if it contributes to the actions of SVE, but as its anatomical location remains unknown, we cannot readily investigate or comment on this possibility.

16. The authors had better to refer this paper (Reebs and Mrosovsky JBR 1989) in the introduction. Reebs demonstrated the effects of wheel running for circadian behavior.

We have now included this reference in the Discussion.

Reviewer #2 (Remarks to the Author):

The authors present a strong and thorough investigation into how circadian rhythms in mice with compromised VIP signalling can be improved through scheduled exercise. This work build on their previous work in this area (Power et al. 2010). In the present paper they replicate the behavioural findings from the previous work, and extend it with a number of in vitro investigations looking at both electrical activity and clock gene expression as assessed with a luciferase reporter. The quality of the work is excellent and the data support their conclusions. I have some suggestions for improving the presentation, but overall the work is solid and of high quality.

Comments:

1. I was initially confused by the use of "~24h", since this is being used in a different

manner than "circadian rhythm" (which by convention in the field is also ~24h). I think given the style of this journal that the definition needs to be repeated in the results section rather than just in the method. The text needs to be sharpened around this. I would recommend highlighting figure 1e first, and then include an analysis showing that when a rhythm is detected in the pre-SVE days, that it is significantly different than that of WT mice. The rationale for the ~24h definition isn't clear. Is this to match them to WT periods, or could it be that this is actually matching the periods of the SVE schedule. Put another way, what would happen if they put these mice on a non24h T cycle of SVE? Would the exercise increase the strength of the network to yield a WT-like period, or would the period match more closely the T-cycle period?

We thank the reviewer for raising this point. We have now re-iterated and clarified what we meant here. We intended to state that following transfer from light-dark conditions, these neuropeptide signaling deficient mice do not spontaneously exhibit rhythms with periods near to that of C57Bl6 wild-type mice (typically ~23.5-23.9h; but objectively defined here as 23.33h-24.34h, based on the mean period of WT mice pre-SVE, plus/minus 2 standard deviations). As noted above, we have now included data from the T-cycle with 25h periodicity of SVE on the *Vipr2*^{-/-} mice – this indicates that most do not sustain ~25h rhythms post-SVE_{25h}, rather that most *Vipr2*^{-/-} mice rhythmic after SVE_{25h} revert to rhythms closer to ~24h.

2. The ephys work would benefit from highlighting in the results section the phase. Some expansion of the rationale for this phase would also be warranted, as CT13 is generally quite late for SCN ephys, with the SCN showing more activity in the mid-subjective day. An investigation of changes of electrical activity across the day would have been interesting, and might be something to consider in follow-up studies. Some of the differences observed could be due to simply sampling non-comparable phases in the various experimental groups, rather than being due to more profound and long term changes in GABA signalling.

As outlined in our response to reviewer 1 point 3, above, we chose to maintain the phase relationship between the time of recording and the presentation of the opportunity to exercise under SVE conditions. We have also added additional comments on this to the results section (lines 239-242). We could not do both experiments (long-term recordings over 24h+ hours and test GABA_A receptors' actions)—this would require a much larger sample size and equipment for higher throughput than was available to us.

3. The data in figure 2 are really exciting as they get around the phase sampling issue mentioned above with respect to the ephys work. It would be helpful if the phase for the photos in panel A was indicated on the traces in panel B.

We have added those details to the panels.

4. Figure 4 is quite nice as well, but thinking back to the analysis in figure 2, one is left wondering if the changes in tissue level rhythms are due to changes in the individual cellular rhythms, or due to changes in phase coherence of the population. Do the authors have data to draw upon, such as in figure 2, to address this?

We think it most likely that this is due to a combination of both possibilities, as suggested by our previous data (Hughes et al 2008), though we here can neither determine this empirically, nor dissect which drives the other or which is more important. As such, we do not speculate on this further. We do not have any further data that can be called upon to address this point.

5. How representative is the trace in the upper left of figure 3d? This seems a lot noisier and more variable than the other traces. I note that the bar in 3C for this trace have more variability than the other bars. This particular trace looks like it would be much higher than ~9Hz. A horizontal line representing the discrimination threshold in each trace might help the reader.

We have added the discrimination threshold bar as suggested to address this.

6. Figure 3g-j represent the experiment with gabazine, although this isn't clear in the figure. The time of application could be indicated in panel 3h, and the word gabazine could appear somewhere in these panels to help the reader. The same can be suggested for figure 4e.

Thank you for raising this, we have now added these.

7. IN the conclusion (and end of the intro) the authors argue that exercise could be used to restore rhythmicity in people with weak clocks such as the elderly. Is there evidence that the VIP/VPAC2 ko mice model similar problems that are observed in the elderly? The authors might also want to consider the interplay between activity and rhythmicity in the elderly. In many cases the lack of activity might be the cause of the weak rhythms. Also those with aging related problems in their rhythms might not be able to exercise. In that sense the work described here investigating the neural mechanisms are critical, as it might help identify interventions that could yield the same effect without requiring exercise itself.

We thank the reviewer for the suggestions and have added a sentence on this to the Discussion.

Lines 481-484. "Indeed, since elderly people can be physically incapacitated and unable to exercise, our findings raise the possibility that drugs activating to reduce GABA signaling may be useful for ameliorating age-related decline in circadian rhythmicity."

Reviewer #3 (Remarks to the Author):

This group has previously reported that *vip*^{-/-} and *vipr2*^{-/-} mice exhibit weak, short tau rhythms in DD that can be entrained to 24h scheduled wheel access, and that show longer taus (aftereffects) in DD following entrainment. In these KO models, scheduled wheel running appears to be a more potent Zeitgeber than LD cycles. The present study confirms these results and describes cellular correlates in the SCN pacemaker, using single cell clock gene imaging and electrophysiology. The results suggest that nonphotic entrainment in the KO mice is facilitated by down-regulation of inhibitory GABA signaling associated with scheduled activity.

results

1. line 109 - The authors state that "mice deficient in VPAC2 (*Vipr2*^{-/-}) or VIP (*Vip*^{-/-}) show apparent daily rhythms under the light-dark cycle but cannot express endogenous ~24h rhythms in behavior when placed in constant dark." The wording is potentially confusing, because the supporting figures clearly show that the KO mice express significant rhythmicity, which could be defined as in the 'circadian range' (e.g., WT mice can entrain to 22h T cycles, which places 22h in the circadian range). This is a minor point, but I would substitute more descriptive wording, and say instead that the KO mice express endogenous rhythms with a periodicity of 22 +xx h, which is significantly shorter than the circadian periodicity of 23 + xx h in WT (and differences in amplitude, precision, etc., whatever else might obtain).

We thank the reviewer for raising this and have altered the text accordingly.

2. I would also avoid saying that the KOs "cannot" express ~24h rhythms, because you then immediately show that they can (i.e., following the wheel access schedules). The KO mice show an aftereffect on tau of entrainment to the 24h activity schedule that is not evident following exposure to 24h LD cycles.

Again, we have altered this text.

3. line 117 - The WT mice did not show an effect of the activity schedule on tau, but the schedule wasn't in place long enough for the WT mice to entrain, so one would not expect to see an aftereffect similar to that evident in the KO mice (aftereffects being a response to entrainment). When the activity schedule is maintained until the WT do entrain, do they show evidence of aftereffects?

We have reported the outcome of this experiment in a previous study (Power et al., 2010) and this effect can also be seen here in Fig. 1b. It requires around 7 weeks of SVE for the WT mice to entrain and show an after-effect of SVE.

4. There were differences between WT and *vipr2*^{-/-} mice in dorsal SCN firing rates. Meaningful comparison is predicated on sampling at the same circadian phase in the two groups. Many of the KO mice did exhibit significant rhythmicity prior to scheduled activity, and thus a CT could have been defined by reference to the regression lines, such as illustrated in Figure 1. In cases where the recordings were made from KO mice with significant rhythmicity, have you tried grouping the recordings according to CT, for comparison with the WT mice, which were sampled at CT9-11?

It is not clear what the reviewer is referring to. The *Vipr2*^{-/-} mice under SVE are not free-running but synchronized to the timed exercise. Therefore, the sampling time is related to the onset of SVE phase.

5. Line 236 - "Coincident with the post-SVE neurophysiological changes in the WT SCN, we recorded a significant reduction in the amplitude of PER2::LUC oscillations following SVE in both doSCN and voSCN from these neuropeptide-competent mice". Were these recordings from WT mice that entrained to SVE, and if not, do you think amplitude would be increased in WT mice culled after entrainment was achieved?

We used strict criteria for inclusion in this study. Yes these WT mice had entrained to SVE, so we can only speculate as to the nature of the amplitude in non-entrained individuals. Previously we observed that the strength of WT behavioral rhythms was diminished following stable entrainment to ~50 days of SVE (Power et al., 2010) and therefore we speculate that subregional SCN PER2::LUC amplitudes would not be increased in WT mice.

6. The results support the authors' conclusion that nonphotic entrainment in the KO mice is facilitated by down-regulation of inhibitory GABA signaling associated with scheduled activity. This is a puzzling effect, given that the manipulated variable is wheel running; if scheduled running downregulates GABA, why does downregulation not occur when the mice have free access to the wheel?

We do not know why spontaneous wheel-running does not feedback to add robustness to the SCN of free-running neuropeptide-signaling deficient mice. We posit that feedback from arousal/exercise does not ordinarily evoke much effect as the single cell oscillators

of the *Vipr2*^{-/-} SCN are somewhat desynchronized and consequently single cell oscillators will receive this feedback information at different phases of their intrinsic circadian cycle. Consequently, this is not permissive for robust re-organization, whereas recurrent feedback from the 24h (and to a certain extent, the 25h) Zeitgeber does achieve this.

Yours sincerely,

Professor Hugh Piggins
Head of School of Physiology, Pharmacology & Neuroscience

Reviewers' comments:

Reviewer #1 (Remarks to the Author):

In the revised manuscript, Hughes et al. provided supplemental analysis and data to address the previous comments. I remain supportive of this work, but there also remain a few issues which I respectfully disagree.

In this research, the authors used Per1-GFP mice because this reporter enable them to measure circadian rhythms in a cellular resolution. Whereas they used PER2::LUC mice because it is possible to record circadian rhythms from the SCN for long range. As the authors explained, unavailability of Per1::Luc mice is understandable. However, several papers suggested that Per1 and Per2 gene show different expression pattern of circadian rhythms in the SCN (Cheng et al 2009 Hum. Mol. Genet.; Riddle et al., 2016 Eur. J. Neurosci.; Yoshikawa et al., 2017 Sci. Rep.). Furthermore, post-transcriptional and translational modification have a potential for modification of circadian rhythms. Thus, the authors need to discuss this issue (Per1 vs. Per2; post-transcription vs. -translation) in the revised manuscript.

I am still confused about the method of MEA recordings. The SCN was cultured at CT9-11 (1-3 hours before the activity onset) from non-SVE control, and 1-3h prior to the scheduled wheel availability from post-SVE (line1134-1137). Fig. S1e shows that the activity onset of WT was located at near ZT12 in LD-DD1, but located around ZT4 in SVE-DD2, indicating that the SCN was cultured around 5-7 hours after the activity onset in post-SVE group. If it is true, timing of brain sampling seemed not the same between non-SVE control and post-SVE control in WT mice.

Figure S2c shows circadian period in behavior during pre- and post-SVE from Fig. 2b. I guess gray and dark bars indicate pre- and post-SVE. But in Fig.S2a and b show only circadian behavior under SVE not including pre-SVE. It would be adequate to include behavior before SVE.

Since Fig. S4b and d show raw data of firing, I recommend to show the raw data in Gabazine application as well.

Also it is better to show PER2::LUC rhythms before and after Gabazine application in Figure 5e.

Reviewer #2 (Remarks to the Author):

The authors have done an excellent job addressing my concerns raised in the initial review. I only have a couple small point for them to address.

1. The issue raised by the other reviewers about studying the electrical activity at comparable phases is an important issue. The authors should address this on page 12 around line 242. This reviewer feels that the best evidence in support of this practice is the phase that the VPAC2^{-/-} free-run from following SVE. The authors state this in the second sentence of their discussion, but it is worth highlighting when this practice is first mentioned to help the skeptical reader understand the rationale.

2. There is a disconnect between figure 1C and figure 1h for the VIP^{-/-} animals. The "representative" trace in 1C isn't very representative. There was no significant change in rhythm strength (1H), but this trace show a very strong rhythm which has a strength of ~40%V that is halved to ~20%V following SVE. Additionally, there does not appear to be an animal in 1H that corresponds to the example presented. There is one animal that has a strength of 45 pre-SVE, and 20 post-SVE (found in the supplied spreadsheet). If that is the animal presented in 1C, then the gray line is not accurate (The peak may not even reach 40 in 1C, while the value in 1H and the spreadsheet is 45). Additionally, this

animal's baseline rhythm strength appears to be an outlier relative to the rest of the animals. One of the other animals would likely be more representative.

Reviewer #3 (Remarks to the Author):

I am satisfied with the response to the reviews.

Response to Reviewers

We thank the reviewers for their comments and continued support for this manuscript. We appreciate that understanding the sampling times for the electrophysiological aspect of the study can be difficult to follow so we have included a new Supplemental Figure S5 to illustrate the design of this facet of the manuscript. Further we have included an additional group of WT animals, SVE(2), for this multi-unit activity investigation, in which SCN electrical activity is sampled with respect to the animals' endogenous rhythm (please see main manuscript file, new Supplemental Fig. S5 and further comments below (response to Reviewer 1, point 2) for details). The parameters of SCN activity that we examined did not vary between this new SVE(2) group and the original SVE(1) group that is sampled with respect to the timing of running-wheel availability. We have also re-analyzed the *Vip*^{-/-} mouse and found an error that we have now recorrected. We have made other minor corrections and additions. We believe that we have addressed concerns raised.

Reviewers' comments:

Reviewer #1 (Remarks to the Author):

In the revised manuscript, Hughes et al. provided supplemental analysis and data to address the previous comments. I remain supportive of this work, but there also remain a few issues which I respectfully disagree.

1. In this research, the authors used Per1-GFP mice because this reporter enable them to measure circadian rhythms in a cellular resolution. Whereas they used PER2::LUC mice because it is possible to record circadian rhythms from the SCN for long range. As the authors explained, unavailability of Per1::Luc mice is understandable. However, several papers suggested that Per1 and Per2 gene show different expression pattern of circadian rhythms in the SCN (Cheng et al 2009 Hum. Mol. Genet.; Riddle et al., 2016 Eur. J. Neurosci.; Yoshikawa et al., 2017 Sci. Rep.). Furthermore, post-transcriptional and translational modification have a potential for modification of circadian rhythms. Thus, the authors need to discuss this issue (Per1 vs. Per2; post-transcription vs. -translation) in the revised manuscript.

Response: We agree that this is an interesting topic, albeit one beyond the scope of this manuscript. Our study was not designed to investigate differential patterns of transcriptional/post-translational modification of *Per1/Per2* so we really cannot comment much on this. We have, however, added a sentence and included citations on this in the final paragraph of the Discussion to indicate that further studies could look at this more closely.

2. I am still confused about the method of MEA recordings. The SCN was cultured at CT9-11 (1-3 hours before the activity onset) from non-SVE control, and 1-3h prior to the scheduled wheel availability from post-SVE (line1134-1137). Fig. S1e shows that the activity onset of WT

was located at near ZT12 in LD-DD1, but located around ZT4 in SVE-DD2, indicating that the SCN was cultured around 5-7 hours after the activity onset in post-SVE group. If it is true, timing of brain sampling seemed not the same between non-SVE control and post-SVE control in WT mice.

Response: We appreciate the concern here and to improve communication of the experimental design, we have included an illustration (new Supplemental Figure S5). The reviewer is correct that because of the different ways that WT and *Vipr2*^{-/-} mice respond and entrain to SVE, there are differences in the sampling times for the electrophysiology. Thus, we have included a new group of WT mice that are sampled at an appropriate circadian time; SVE(2) group. We now, therefore, present data for groups of WT mice culled/recorded at a time to control for the time on onset of endogenous activity and separately for the time of wheel-running activity. However, there are no differences in the spontaneous firing rate in the dSCN and vSCN of this SVE(2) group and that of the original WT SVE group (now labelled SVE(1); Figure 4). Further, actions of gabazine are not different between the SVE(1) and SVE(2) groups (Figure 5). Supplemental Tables 1 and 2 summarize these statistics. Thus, although sampled at different times, these parameters of WT SCN activity do not vary.

3. Figure S2c shows circadian period in behavior during pre- and post-SVE from Fig. 2b. I guess gray and dark bars indicate pre- and post-SVE. But in Fig.S2a and b show only circadian behavior under SVE not including pre-SVE. It would be adequate to include behavior before SVE.

Response: Apologies if this was not clearer. The actograms shown in supplemental Fig. S2a and S2b are not the data shown in supplemental Figure S2c. Supplemental Figure S2c shows the data from Figure 2b (NOT SUPPLEMENTAL Figure S2b) as not all data were included in Figure 2b. The reason for that is this was a repeated measures design and some animals could not be analyzed as they either had no measurable rhythm pre-SVE or no measurable rhythm post-SVE. This supplemental figure S2c is just included for completion sake as it shows all datapoints, regardless of whether animals were rhythmic both pre- and post-SVE or not. The colour coding is pink for pre-SVE and red for post-SVE in figure 2b, and pink pre-SVE with red post-SVE in Figure S2c. We have now added a legend to figure S2c to clarify this.

4. Since Fig. S4b and d show raw data of firing, I recommend to show the raw data in Gabazine application as well.

Response: We have included raw data of firing prior to and during gabazine treatment in Figure 5. Note that the time scalebar for the raw data examples shown in Figure 5 is 10s (to capture firing pre- and during gabazine), whereas in Figure 4 we are using a 500ms scalebar to illustrate the spontaneous firing rate.

5. Also it is better to show PER2::LUC rhythms before and after Gabazine application in Figure 5e.

Response: We respectfully wish to leave the figure as is; the differential effects of gabazine on PER2::LUC are clear in the conditions/genotypes. Exemplar rhythms in PER2::LUC can be visualized in the upper panels (a-d) of Figure 6, while below are examples of PER2::LUC rhythms from a control non-SVE *Vipr2*^{-/-} SCN and SVE *Vipr2*^{-/-} SCN, prior to and following gabazine treatment (indicated by the arrow)

Reviewer #2 (Remarks to the Author):

The authors have done an excellent job addressing my concerns raised in the initial review. I only have a couple small point for them to address.

1. The issue raised by the other reviewers about studying the electrical activity at comparable phases is an important issue. The authors should address this on page 12 around line 242. This reviewer feels that the best evidence in support of this practice is the phase that the VPAC2^{-/-} free-run from following SVE. The authors state this in the second sentence of their discussion, but it is worth highlighting when this practice is first mentioned to help the skeptical reader understand the rationale.

Response: We have included the additional SVE(2) group for WT mice and the parameters of SCN neurophysiology that we measured did not differ to the SVE(1) group (please see above in our response to Reviewer 1).

2. There is a disconnect between figure 1C and figure 1h for the VIP^{-/-} animals. The "representative" trace in 1C isn't very representative. There was no significant change in rhythm strength (1H), but this trace show a very strong rhythm which has a strength of ~40%V that is halved to ~20%V following SVE. Additionally, there does not appear to be an animal in 1H that corresponds to the example presented. There is one animal that has a strength of 45 pre-SVE, and 20 post-SVE (found in the supplied spreadsheet). If that is the animal presented in 1C, then the gray line is not accurate (The peak may not even reach 40 in 1C,

while the value in 1H and the spreadsheet is 45). Additionally, this animal's baseline rhythm strength appears to be an outlier relative to the rest of the animals. One of the other animals would likely be more representative.

Response: We thank the reviewer for bringing this to our attention. The *Vip*^{-/-} rhythm strength data point mentioned at ~45%V was in fact an error that we think arose when data files were updated after periodogram analyses were re-run in Clocklab (as a precaution, we have now re-checked data from all genotypes). We have now corrected the graphs shown in Figure 1h and Supplementary Figure S1h, as well as the *Vip*^{-/-} rhythm strength summary data in Table 1, and the data tables submitted with the revised manuscript. This data point should sit at ~38%V rather than ~45%V. As such, we agree that this example is still at the high end of rhythm strength pre-SVE for *Vip*^{-/-} mice, but we think it remains the best example to illustrate the period change from ~22h to ~24h, which we feel is the most important information that should be clearly visible in the examples shown in the figure.

Reviewer #3 (Remarks to the Author):

I am satisfied with the response to the review

REVIEWERS' COMMENTS:

Reviewer #1 (Remarks to the Author):

The authors have responded satisfactorily to my comments. I only have one comment.

It would be easier to understand if the authors add the PER2::LUC data before and after Gabazine application in corresponding to Figure 6i.

Reviewer #2 (Remarks to the Author):

The revised manuscript has expertly addressed all of my concerns.

Response to Reviewers

We thank the reviewers for their comments and continued support for this manuscript. To address Reviewer 1's concern about the PER2::LUC traces and effects of gabazine, we have added four traces to what was Figure 6i. This is now Figure 7 as there was not enough space in the original Figure 6 to accommodate these additional traces. We believe that we have addressed concerns raised.

Reviewer #1 (Remarks to the Author):

The authors have responded satisfactorily to my comments. I only have one comment.

It would be easier to understand if the authors add the PER2::LUC data before and after Gabazine application in corresponding to Figure 6i.

Response: We have included 4 traces illustrating how PER2::LUC rhythms respond to gabazine treatment. These were added to what was Figure 6i. However, as there was not enough space in Figure 6i, we created a new figure, Figure 7, which contains the original Figure 6i and the new 4 traces.